# lncRNA LINC00941 modulates MTA2/NuRD occupancy to suppress premature human epidermal differentiation

Eva Morgenstern[1] , Carolin Molthof[1], Uwe Schwartz[2] , Johannes Graf[1], Astrid Bruckmann[1] , Sonja Hombach[1,3], Markus Kretz[1,3]

**Numerous long non-coding RNAs (lncRNAs) were shown to have a functional impact on cellular processes such as human epidermal homeostasis. However, the mechanism of action for many lncRNAs remains unclear to date. Here, we report that lncRNA LINC00941 regulates keratinocyte differentiation on an epigenetic level through association with the NuRD complex, one of the major chromatin remodelers in cells. We find that LINC00941 interacts with NuRD-associated MTA2 and CHD4 in human primary keratinocytes. LINC00941 perturbation changes MTA2/NuRD occupancy at bivalent chromatin domains in close proximity to transcriptional regulator genes, including the *EGR3* gene coding for a transcription factor regulating epidermal differentiation. Notably, LINC00941 depletion resulted in reduced NuRD occupancy at the *EGR3* gene locus, increased EGR3 expression in human primary keratinocytes, and increased abundance of EGR3-regulated epidermal differentiation genes in cells and human organotypic epidermal tissues. Our results therefore indicate a role of LINC00941/NuRD in repressing EGR3 expression in non-differentiated keratinocytes, consequentially preventing premature differentiation of human epidermal tissues.**

## Introduction

One of the important findings of high-throughput sequencing was that the transcriptional landscape is more complex than originally imagined: it could be shown that only a diminutive fraction of the human genome (<2%) encodes proteins, whereas more than 75% is actively transcribed into non-coding RNAs (ncRNAs), including long non-coding RNAs (lncRNAs) (Djebali et al, 2012). lncRNAs are defined as a class of RNAs longer than 200 nucleotides with no or hardly any protein-coding potential (Bonasio & Shiekhattar, 2014). Most lncRNAs are – similar to mRNAs – transcribed by RNA polymerase II, often capped by 7-methylguanosine ($m^7G$) at their 5′ ends, 3′ polyadenylated, and spliced (Quinn & Chang, 2016). Many

lncRNAs have been shown to be crucial for numerous biological processes, such as X chromosome inactivation (Strehle & Guttman, 2020), epigenetic control of chromatin (Kazimierczyk & Wrzesinski, 2021), and imprinting (Sanli et al, 2018).

The most fundamental function of the human skin and the epidermis, its outermost layer, is to provide a barrier to the external environment. The human epidermis is a stratified surface epithelium constantly renewing approximately every 4 wk. This regenerative capacity is mostly maintained by progenitor keratinocytes located in the epidermal basal layer. A subset of daughter cells generated by these progenitor cells dissociate from the basement membrane and migrate to the apical part of the tissue while undergoing a terminal differentiation program. To ensure the formation of a functional epidermal barrier, a precise balance between progenitor and differentiated keratinocytes is crucial (Blanpain & Fuchs, 2006, 2009). Several lncRNAs were previously shown to have functional roles in normal epidermal homeostasis including ANCR, TINCR, and SMRT-2 (Kretz et al, 2012, 2013; Lee et al, 2018). Another lncRNA involved in this process is LINC00941, which was demonstrated to be enriched in undifferentiated progenitor keratinocytes and functioned as a repressor of keratinocyte differentiation (Ziegler et al, 2019). The mechanism by which LINC00941 prevents the onset of epidermal differentiation has not been investigated yet.

Here, we show that lncRNA LINC00941 binds several components of the nucleosome remodeling and deacetylase (NuRD) complex, including MTA2, one of its core subunits. Through interaction with NuRD-associated MTA2, LINC00941 modulates occupancy of the epigenetic regulator to mainly transcriptionally repressed and bivalent chromatin sites, thus helping to regulate its gene regulatory function. Chromatin immunoprecipitation (ChIP) and RNA-sequencing approaches showed LINC00941/MTA2-mediated repression of EGR3, a transcription factor responsible for keratinocyte differentiation (Kim et al, 2019). Thus, LINC00941 prevents premature differentiation by interacting with the MTA2/NuRD complex inhibiting transcription of early and late differentiation genes through their upstream regulator EGR3.

[1]Regensburg Center for Biochemistry (RCB), University of Regensburg, Regensburg, Germany  [2]NGS Analysis Center Biology and Pre-Clinical Medicine, University of Regensburg, Regensburg, Germany  [3]Institute for Molecular Medicine, MSH Medical School Hamburg, Hamburg, Germany

Correspondence: markus.kretz@medicalschool-hamburg.de

# Results

## The lncRNA LINC00941 interacts with components of the NuRD complex

LINC00941, a lncRNA with partial nuclear localization, was recently shown to regulate the expression of early and late key differentiation genes in human epidermis. These included most of the genes within the small proline-rich (SPRR) protein and late cornified envelope (LCE) protein clusters located in the epidermal differentiation complex (EDC) (Ziegler et al, 2019). Based on these findings, we hypothesized a LINC00941-mediated regulation of gene expression on a global level through an epigenetic mechanism. To test this hypothesis and to get a deeper insight into the mode of action of LINC00941 in human epidermal homeostasis, we performed RNA–protein interactome analyses to identify proteins associated with LINC00941. For this, RNA pull-down and mass spectrometry (MS) analyses with in vitro–transcribed, biotinylated LINC00941, as well as cell lysates from primary keratinocytes, were performed. Correspondingly, we found that LINC00941 interacts with proteins attributed to chromatin-associated functions, such as nucleosomal DNA binding or histone deacetylase binding properties, as shown by chromatin-associated Gene Ontology (GO) term analysis (Fig 1A). Interestingly, LINC00941-bound proteins were enriched for components of chromatin remodeling complexes – in particular, the NuRD complex. A more detailed analysis of the NuRD components identified with our analysis showed that LINC00941 interacted with CHD4, MTA2, HDAC2, GATAD2A, GATA2DB, RBBP4, and RBBP7, respectively (Figs 1B and S1A; Table S1). The NuRD complex combines both ATP-dependent chromatin remodeling and histone deacetylase activities modulating gene expression. Interestingly, some components of the NuRD complex such as HDAC1/2, CHD4, and MTA2 have previously been demonstrated to fulfill essential functions in murine epidermal development (Kashiwagi et al, 2007; Lu et al, 2008; Leboeuf et al, 2010). Therefore, we hypothesized that NuRD-mediated epigenetic regulation of differentiation gene expression— including genes in the EDC — might be at least in part controlled by LINC00941. To test our hypothesis, the interaction between LINC00941 and the NuRD complex was subsequently further verified. Because multiple components of the NuRD complex such as RBBP4, RBBP7, and HDAC2 can also function as subunits of other multiprotein complexes, including Sin3, Extra Sex Combs/Enhancer of Zeste (ESC/E[Z]), and nucleosome remodeling factor (NURF) complex (Dubey et al, 2017; Pantier et al, 2017; Zahid et al, 2021), we focused our experiments on MTA2, a core subunit of the NuRD complex (Low et al, 2020). The binding of in vitro–transcribed, biotinylated LINC00941 and NuRD-associated MTA2 from a cell lysate with subsequent RNA pull-down was confirmed by Western blot analysis (Fig 1C). Correspondingly, RNA immunoprecipitation (RNA-IP) and subsequent qRT–PCR analysis also verified an interaction between overexpressed LINC00941 and endogenous MTA2, as well as CHD4 in primary keratinocytes (Figs 1D and S1B), strongly indicating an association of LINC00941 with the chromatin remodeling complex NuRD. This finding suggested a role of the lncRNA in regulating epidermal homeostasis through modulation of NuRD activity or recruitment.

## NuRD-associated MTA2 prevents premature differentiation of human keratinocytes

We recently reported that LINC00941 was highly induced in undifferentiated progenitor keratinocytes and reduced in abundance as keratinocyte differentiation progressed (Ziegler et al, 2019). To test whether NuRD-associated MTA2 shows a similar dynamic regulation during human keratinocyte differentiation, we performed qRT–PCR analysis of MTA2 throughout six time points of calcium-induced human keratinocyte differentiation. We found that MTA2 mRNA was highly abundant in non- and poorly differentiated keratinocytes but repressed during keratinocyte differentiation, thus exhibiting an expression pattern similar to LINC00941 (Fig 2A). Correspondingly, Western blot analyses with MTA2-specific antibodies also confirmed declining protein amounts of MTA2 in calcium-induced differentiated keratinocytes (Fig 2B). Similar to MTA2, mRNA and protein levels of CHD4 — another core component of the NuRD complex — decreased in the course of keratinocyte differentiation (Fig 2A and B). These results of correlating expression patterns further suggested a functional interaction between the LINC00941 and the NuRD complex.

To test whether MTA2 affects human epidermal homeostasis in a similar manner as LINC00941, we generated MTA2-deficient organotypic human epidermis using pools of 11–30 siRNAs to achieve efficient and specific depletion of MTA2 mRNA and proteins (Fig S2A). MTA2-deficient organotypic epidermis showed increased mRNA abundance of the early differentiation gene *keratin 1* and the late differentiation gene *filaggrin* (Fig 2C). Correspondingly, LINC00941-deficient organotypic epidermis yielded similarly increased abundance of both mRNA populations (Fig 2D). Therefore, both LINC00941- or MTA2-deficient tissues showed premature and more advanced differentiation compared with control-treated organotypic epidermis, as indicated by increased protein abundance of keratin 10, an early differentiation marker co-expressed with keratin 1, and the late differentiation marker filaggrin (Fig 2E). Interestingly, knockdown of LINC00941 and MTA2, respectively, resulted in significantly decreased mRNA abundance of other NuRD components, including MTA1, MTA3, CHD3, and CHD4, thus suggesting impairment of NuRD complex formation upon knockdown of LINC00941 or associated MTA2 (Fig S2B and C). Together, the above results suggested a common role of the complex consisting of LINC00941 and NuRD in repressing a premature onset of keratinocyte differentiation in human epidermal tissues.

## MTA2/NuRD occupies regulatory regions in keratinocytes

Because LINC00941 interacted with NuRD-associated MTA2 and both appeared to have overlapping functionality, we hypothesized a role of LINC00941 in modulating NuRD-mediated epigenetic regulation of genes relevant for epidermal homeostasis. To examine the genome-wide impact of LINC00941 deficiency on NuRD binding to chromatin, we performed ChIP sequencing with LINC00941-deficient and control-treated keratinocytes using antibodies directed against MTA2. First, we wanted to get a deeper understanding of the general chromatin binding behavior of the NuRD complex in keratinocytes and therefore identified genome-wide MTA2/NuRD binding sites ($n$ = 3,613) (Table S2), which were

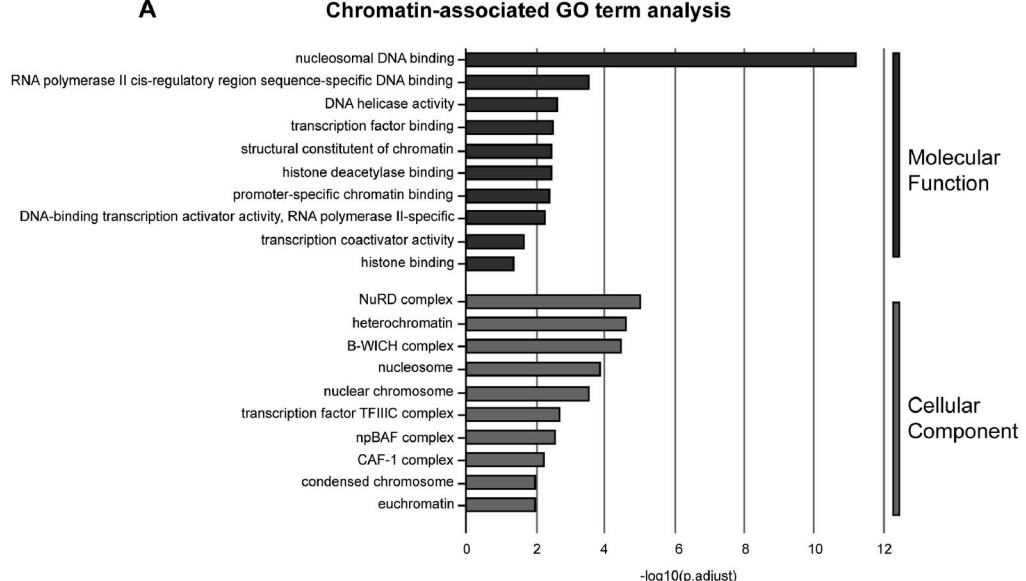

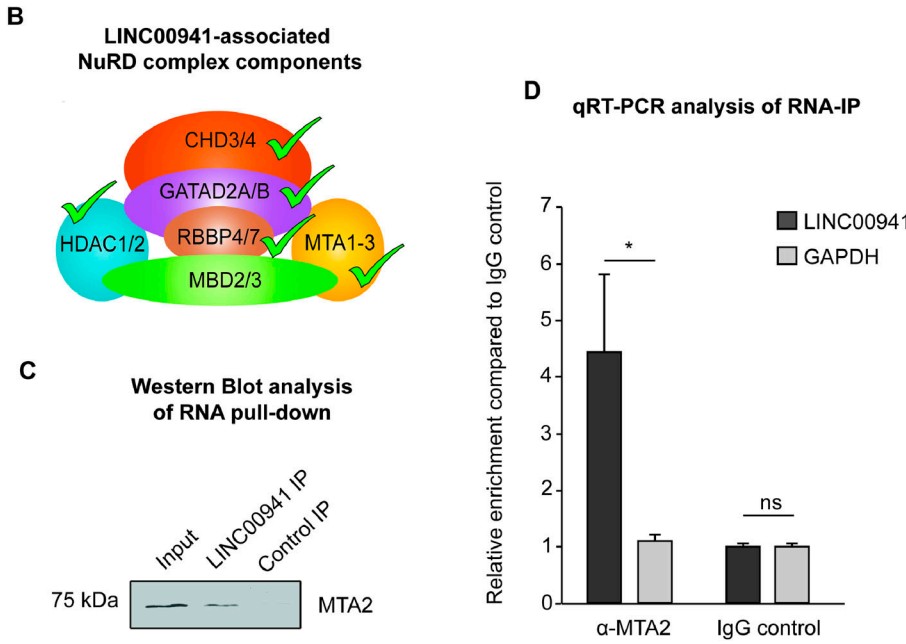

**Figure 1. LINC00941 interacts with components of the NuRD complex.**
**(A)** Chromatin-associated Gene Ontology (GO) terms of LINC00941-bound proteins obtained by mass spectrometry (MS) analysis. The NuRD complex showed the strongest signal with regard to the cellular component. **(B)** Scheme of NuRD components found to interact with lncRNA LINC00941 in MS analysis. **(C)** RNA pull-down with in vitro–transcribed and biotinylated LINC00941 showed interaction with MTA2 in Western blot analysis. **(D)** RNA immunoprecipitation (RNA-IP) with overexpressed LINC00941 verified interaction between LINC00941 and MTA2 by qRT–PCR ($n = 3$). Data are presented as ± SD. Statistical significance was tested by an unpaired $t$ test (*adj. $P < 0.05$ and ns, not significant).

highly reproducible between replicates (Fig S3A). We observed enrichment of MTA2/NuRD occupancy at promoters (Figs 3A and S3B). GO term analysis of NuRD-associated promoters included genes regulating cell fate commitment, as well as morphogenic and developmental processes (Figs 3B and S3C). Importantly, these terms also comprise targets important for keratinocyte development and differentiation such as *MAFB*, *EPHA2*, and *SOX9*.

To analyze the chromatin context at MTA2/NuRD occupied sites, we used the epigenome roadmap chromatin state model of primary keratinocytes (Roadmap Epigenomics Consortium et al, 2015) and found that NuRD-associated MTA2 bound to various chromatin states at regulatory elements (Fig 3C). MTA2/NuRD was associated not only with activated chromatin states such as active transcription start sites (TssA, TssAFlnk) and enhancers (Enh, EnhG) but

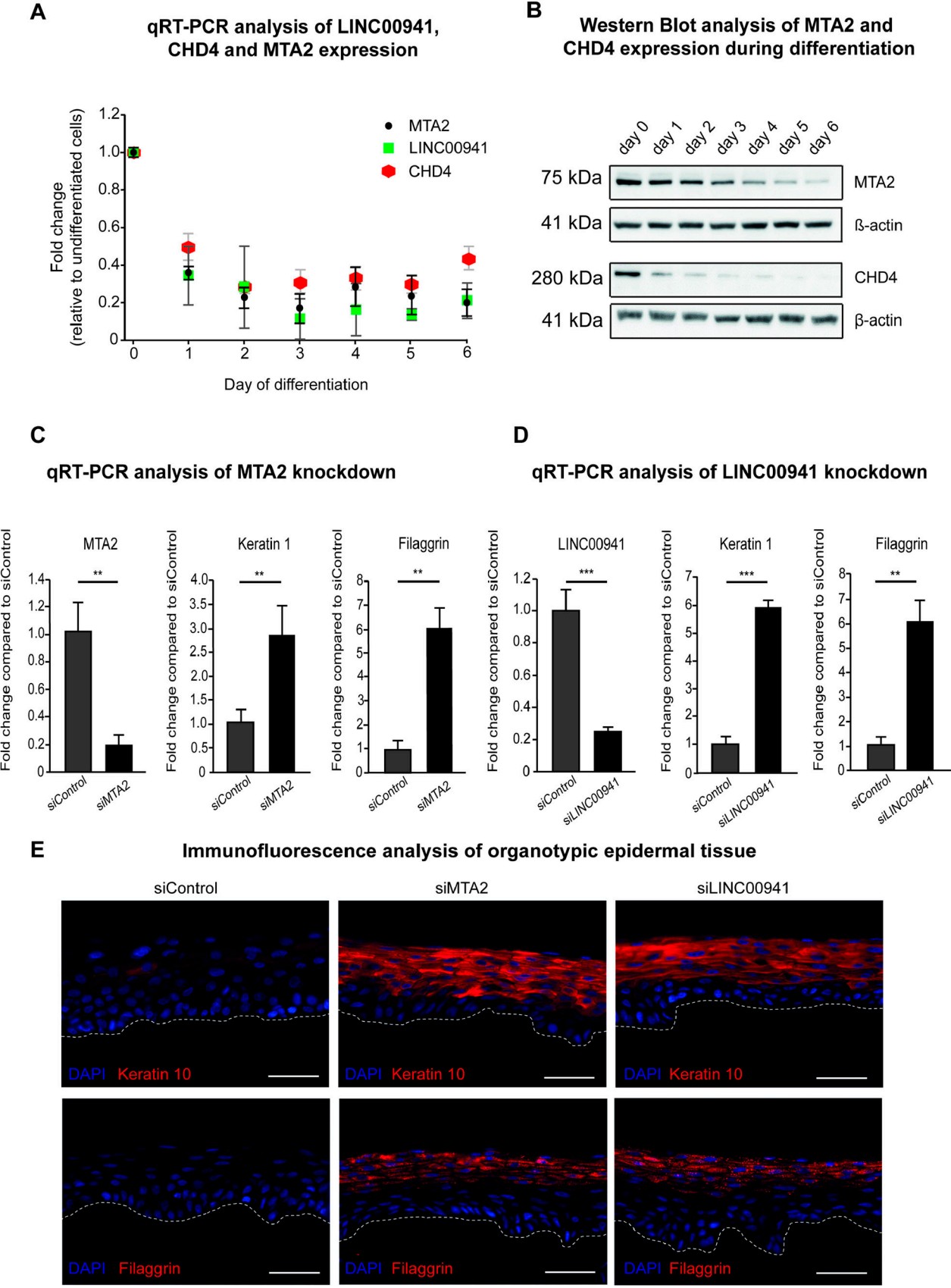

**A** qRT-PCR analysis of LINC00941, CHD4 and MTA2 expression

**B** Western Blot analysis of MTA2 and CHD4 expression during differentiation

**C** qRT-PCR analysis of MTA2 knockdown

**D** qRT-PCR analysis of LINC00941 knockdown

**E** Immunofluorescence analysis of organotypic epidermal tissue

also with repressed chromatin states (ReprPC). Furthermore, it was also frequently bound to bivalent chromatin states including bivalent transcription start sites, enhancers, and related flanking regions (TssBiv, BivFlnk, and EnhBiv) (Fig 3D). In total, of 7,653 genes annotated with bivalent promoter regions, 1,085 genes are bound by MTA2 in human keratinocytes (Fig S3D). A more detailed analysis of some representatives of those different chromatin states confirmed the versatile chromatin binding behavior of MTA2/NuRD: ESYT3, a membrane-associated protein predicted to bind calcium ions and phospholipids, represented one example of a bivalent MTA2/NuRD occupancy site. It harbored both active (H3K4me3, H3K27ac, and H3K4me1) and repressive (H3K27me3) histone marks at the MTA2 occupied site, resulting in the sparse expression of the target gene. The *PCDHGC3* gene locus represents a typical example of MTA2/NuRD occupancy within activated chromatin sites (TssA, TssAFlnk). Correspondingly, solely activating histone marks could be detected (Fig 3E). PCDHGC3 is an adhesion protein, potentially calcium-dependent and expressed in human keratinocytes. The broad range of MTA2/NuRD-associated chromatin states in human keratinocytes emphasized a complex role of NuRD as an epigenetic regulator complex — beyond the role as a transcriptional repressor (Miccio et al, 2010; Hu & Wade, 2012; Pundhir et al, 2023). Interestingly, its distribution of differential peaks at genomic features of primary keratinocytes showed a stronger preference for promoter regions opposite to several reports published previously, which detected a preference for intragenic regions and a subordinate role of promoters (Arends et al, 2019; Lu et al, 2019; Marques et al, 2020). However, ChIP-sequencing analysis of NuRD-associated MBD3 in mouse embryonic stem cells found similar enrichment of the NuRD complex at promoter regions, generally indicating a cell type– and cell state–dependent occupancy pattern (Yildirim et al, 2011).

## LINC00941 dependency of MTA2/NuRD binding at *EGR3*

Next, we tested whether the binding behavior of MTA2/NuRD changes in LINC00941-deficient cells. Unsupervised principal component analysis and scatterplots revealed that the highest variance in MTA2/NuRD occupancy across all samples could be explained by the siRNA-mediated LINC00941 knockdown, thus suggesting a regulatory role of LINC00941 in MTA2/NuRD chromatin binding (Figs 4A and S4A). Even though LINC00941 does not globally influence MTA2 chromatin binding, differential occupancy analysis resulted in 33 significantly altered MTA2/NuRD binding sites upon LINC00941 depletion (false discovery rate [FDR] threshold of 5%) (Figs 4B and S4B and C and Table S3). 67% of all differential MTA2/NuRD occupied sites were located directly at the TSS, suggesting

LINC00941 as a putative regulator of transcription through MTA2/NuRD (Fig S4D). Most of the differential MTA2/NuRD occupied sites (29 of 33) were marked by the repressive histone modification H3K27me3 corresponding to Polycomb-repressed and bivalent chromatin states (Fig S4E), which is in agreement with the known function of NuRD to enhance PRC2 binding (Kim et al, 2015). In line with that, most of the associated genes were not or sparsely transcribed in undifferentiated keratinocytes (Fig 4B). Most of the 29 target sites showed significantly reduced occupancy of MTA2/NuRD in LINC00941 knockdown cells. In these cases, LINC00941 appeared to facilitate NuRD complex binding to the target genes, thereby promoting transcriptional repression by PRC2. Interestingly, four binding sites were associated with active chromatin states and their corresponding target genes (*AP3D1*, *MROH6*, *SLC25A45*, and *C2CD2*) were considerably expressed in undifferentiated keratinocytes. Here, LINC00941 deficiency resulted in enhanced MTA2/NuRD binding, indicating that LINC00941 bound to the MTA2/NuRD complex appeared to diminish the binding of the NuRD complex to these target genes and ensure their transcription. Furthermore, many of the genes associated with altered MTA2/NuRD occupancy in LINC00941-deficient keratinocytes were transcriptional regulators, some of which were known to be involved in differentiation processes. Therefore, we hypothesized that LINC00941/NuRD might suppress the expression of transcriptional activators. This repression would be restricted to non- and weakly differentiated strata of the epidermis, because the levels of LINC00941 and MTA2, respectively, declined during differentiation (Fig 2A). To verify this hypothesis, we analyzed to what extent MTA2/NuRD-associated genes dynamically changed their expression during keratinocyte differentiation and whether their transcription is influenced by LINC00941 knockdown (Fig 4C). In addition to a solute carrier protein (SLC25A45) (Fig S4F), only the transcription factor EGR3 was fulfilling both criteria: *EGR3* showed both significant up-regulation upon LINC00941 knockdown in organotypic epidermal tissues and up-regulation during keratinocyte differentiation. In addition, ChIP-qPCR analyses (Fig 4E) confirmed the reduced occupancy of MTA2/NuRD at the *EGR3* locus in LINC00941 knockdown cells observed in our ChIP-sequencing experiments. Interestingly, an analysis of the chromatin states at the *EGR3* gene locus revealed that it was annotated as a bivalent and repressed site up to and including day 2.5 of calcium-induced differentiated keratinocytes (Fig 4D). No later than day 5.5, this site was associated with an active chromatin state. These findings were also supported by ChIP-sequencing data of histone modifications and RNA-sequencing data of calcium-induced differentiated keratinocytes from ENCODE. Up to day 2.5 of differentiation, repressive H3K27me3 was the most prevalent

**Figure 2. NuRD-associated MTA2 prevents premature onset of keratinocyte differentiation.**
**(A)** qRT–PCR analysis of LINC00941, MTA2, and CHD4 repression during calcium-induced differentiation in primary keratinocytes compared with undifferentiated keratinocytes (day 0) (*n* = 3–4). Data are presented as the mean ± SD. **(B)** Western blot analysis of MTA2 and CHD4 repression during calcium-induced differentiation in primary keratinocytes (days 1–6) compared with undifferentiated keratinocytes (day 0). **(C, D)** qRT–PCR analysis of either MTA2 or LINC00941 knockdown. **(C, D)** siPool-mediated knockdown of MTA2 (C) and LINC00941 (D), respectively, resulted in increased abundance of early and late differentiation markers keratin 1 and filaggrin on day 3 of differentiation in organotypic epidermal tissues (*n* = 3–5 tissue cultures/knockdown group). Data are presented as the mean ± SD. Statistical significance was tested by an unpaired *t* test and corrected for multiple testing after Bonferroni (***adj. *P* < 0.001 and **adj. *P* < 0.01). **(E)** Immunofluorescence (IF) analysis showed an increased level of early and late differentiation proteins keratin 10 and filaggrin. The dashed line indicates the basement membrane, nuclei are shown in blue, and the differentiation proteins keratin 10 and filaggrin are shown in red (*n* = 3–5 tissue cultures/knockdown group, day 3 of differentiation, one exemplary picture for each group is depicted). Scale bar: 100 *μ*m.

**A**

### Genomic region annotation of MTA2 binding sites

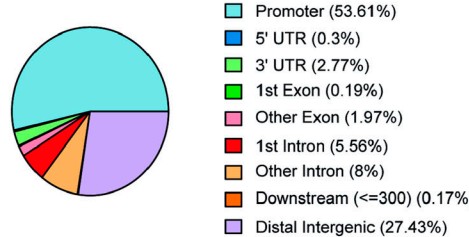

- Promoter (53.61%)
- 5' UTR (0.3%)
- 3' UTR (2.77%)
- 1st Exon (0.19%)
- Other Exon (1.97%)
- 1st Intron (5.56%)
- Other Intron (8%)
- Downstream (<=300) (0.17%)
- Distal Intergenic (27.43%)

**B**

### Top GO terms of MTA2 ChIP sequencing

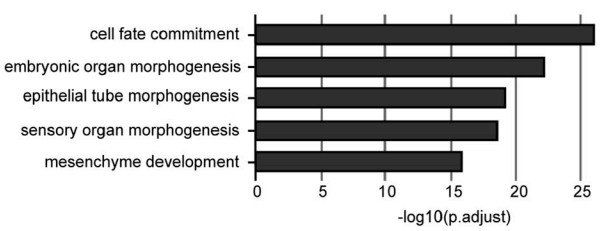

**C**

### Chromatin states at MTA2 binding sites

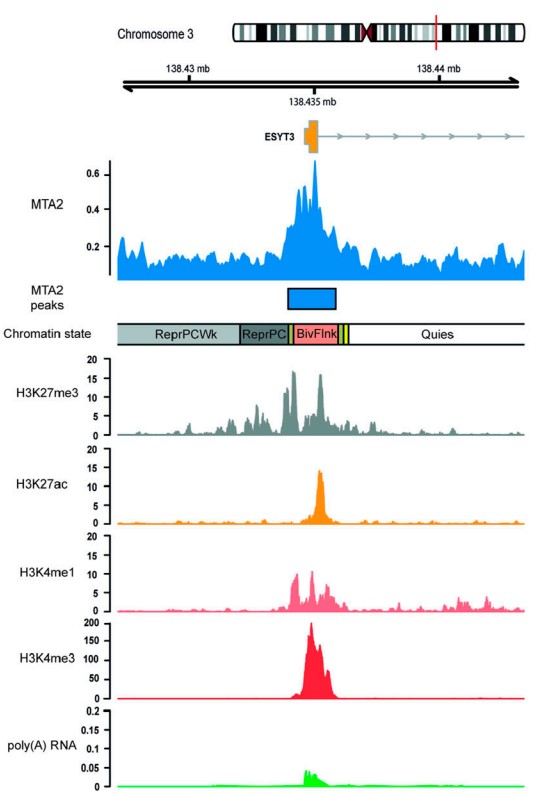

**D**

### MTA2 binds bivalent chromatin sites

**E**

### MTA2 binds active chromatin sites

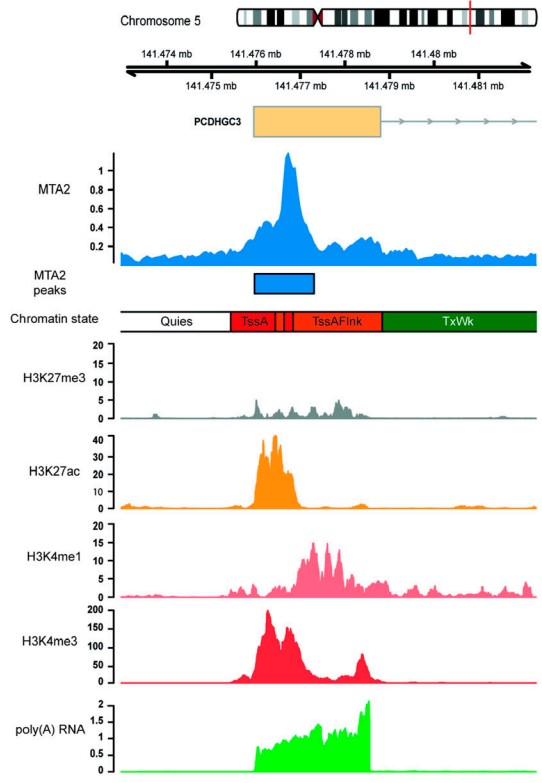

mark in undifferentiated keratinocytes but was erased in the course of differentiation. On the contrary, the transcription-activating chromatin modification H3K27ac increased over time. Correspondingly, the first *EGR3* transcripts were detectable at day 5.5 based on RNA-sequencing data in wild-type keratinocytes (Fig 4C and D). This led us to hypothesize that LINC00941/NuRD binds to the *EGR3* gene locus during repressed and bivalent states, respectively, promoting transcriptional repression. In the course of the differentiation process, the chromatin states at the *EGR3* locus changed and LINC00941 and MTA2 were transcriptionally repressed themselves (Fig 2A), thus allowing for active transcription of *EGR3*. In accordance with this hypothesis, LINC00941 knockdown led to a premature increase in EGR3 abundance in organotypic epidermis, thus accelerating this tightly regulated process (Fig 4F). In conclusion, our data strongly suggested a LINC00941-regulated, epigenetic repression of the *EGR3* gene locus by MTA2/NuRD, therefore preventing premature transcriptional activity during epidermal homeostasis.

### EGR3 promotes keratinocyte differentiation

EGR3 was recently described as a regulator of keratinocyte differentiation (Kim et al, 2019). Correspondingly, we found that EGR3 protein abundance was low in undifferentiated, primary human keratinocytes and strongly induced during calcium-induced differentiation (Fig 5A). In addition to differentiation genes, Kim et al identified 20 functional EGR3-regulated genes, which showed increased expression in calcium-induced differentiated keratinocytes and reduced abundance upon EGR3 knockdown, and which were found to be co-expressed with EGR3. To further support our hypothesis that LINC00941-dependent decrease in NuRD binding at *EGR3* affected its function, we performed an overlap between RNA-sequencing data from LINC00941-deficient cells and EGR3-regulated genes found by Kim et al. Interestingly, all EGR3-regulated genes were increased in abundance in LINC00941-depleted, organotypic epidermal tissues even though they are normally not expressed until day 5.5 in calcium-induced differentiated keratinocytes (Figs 5B and S5A). To verify the functional impact of EGR3 in our organotypic model for human epidermal homeostasis, we generated organotypic epidermal tissues with EGR3-deficient, as well as siLINC00941- and siControl-treated, keratinocytes (Fig 5C and D). Knockdown of EGR3 led to a decrease in EGR3 protein abundance, whereas LINC00941 deficiency resulted in an increase in EGR3 protein abundance — in a similar fashion as previously seen on the RNA level (Fig S5B). We were able to detect significantly reduced mRNA levels of keratin 1 and

filaggrin, respectively. In addition, early and late differentiation markers keratin 10 and filaggrin showed severely reduced abundance in EGR3-deficient organotypic epidermis. We also detected severely reduced abundance of LINC00941-regulated differentiation genes located within the EDC cluster as a consequence of EGR3 knockdown (Fig S5C). These results indicated an important role of EGR3 in promoting keratinocyte differentiation in human epidermal tissues through reduced abundance of EDC-associated not only late but also early differentiation genes. To further test functional interdependency between LINC00941/NuRD and EGR3, we generated organotypic epidermal tissues with keratinocytes simultaneously treated with EGR3 and LINC00941 siRNA pools. Interestingly, we observed no reduction, but a moderate increase in EGR3 mRNA abundance in siEGR3 + siLINC00941 double-treated tissues compared with controls. Correspondingly, abundance of differentiation mRNAs filaggrin and LCE1E was moderately increased, suggesting increased EGR3 transcription because of the lack of LINC00941/NuRD-mediated repression (Fig S5D).

In conclusion, our data strongly suggest a role of lncRNA LINC00941 in regulating recruitment of the chromatin remodeling complex NuRD to target gene loci, thus controlling epidermal homeostasis. Correspondingly, LINC00941/NuRD-dependent epigenetic repression of the transcription factor EGR3 — a regulator of epidermal differentiation — represents a lncRNA-dependent regulatory mechanism for modulation of the epidermal homeostatic balance.

## Discussion

Previous work identified LINC00941 as a negative regulator of premature human keratinocyte differentiation, but the exact mechanism remained unknown (Ziegler et al, 2019). LINC00941-mediated regulation of many genes within the SPRR protein, as well as the LCE protein clusters located within the EDC, suggested a role of LINC00941 as an epigenetic regulator of transcription (Ziegler et al, 2019; Morgenstern & Kretz, 2023). The above study has confirmed this hypothesis and unraveled the mode of action of LINC00941 as a transcriptional modulator of epidermal homeostasis for the first time. Even though LINC00941 does not globally influence MTA2 binding, this study demonstrated LINC00941-dependent MTA2/NuRD binding at several chromatin sites, including *EGR3*, a transcriptional regulator of keratinocyte differentiation (Fig 4B). Consequentially, LINC00941 might act as a specific modulator of MTA2 chromatin binding to a subset of MTA2 binding

**Figure 3. MTA2/NuRD occupies regulatory regions in keratinocytes.**
**(A)** Pie chart showing genomic region annotation of MTA2 binding sites, preferentially binding to promoter and distal intergenic regions. The genome annotation of protein-coding genes was used. The promoter region was defined as ±1,000-bp distance to the TSS. **(B)** Top GO terms at MTA2 binding sites. The respective GO terms also included genes involved in keratinocyte differentiation. GO term analysis was restricted to terms associated with a minimum of 250 and a maximum of 350 genes. **(C)** Bar plots showing the distribution of primary keratinocyte chromatin states at MTA2 binding sites. The left bar represents the distribution of chromatin states over the whole genome and the right bar the MTA2 binding sites. The enrichment of the chromatin states at MTA2 binding sites is shown as the $\log_{10}$ fold change of the respective chromatin state against the whole genome distribution. MTA2 was found to bind activated (TssA, TssAFlnk, Enh, EnhG), repressed (ReprPC), and bivalent (TssBiv, EnhBiv, BivFlnk) chromatin states. Abbreviations: Tss, transcription start site; A, active; Flnk, flanking; Biv, bivalent; Tx, transcription; Wk, weak; Enh, enhancer; G, genic; Rpts, repeats; Het, heterochromatin; Repr, repressed; PC, Polycomb; Quies, quiescent. **(D, E)** Genome browser view of MTA2 binding sites at selected genomic regions. Tracks of chromatin states, histone modifications, and transcription in primary keratinocytes obtained from roadmap (accession: E057) are shown below the MTA2 ChIP-sequencing tracks. Abbreviations: Tss, transcription start site; A, active; Flnk, flanking; Biv, bivalent; Tx, transcription; Wk, weak; Quies, quiescent.

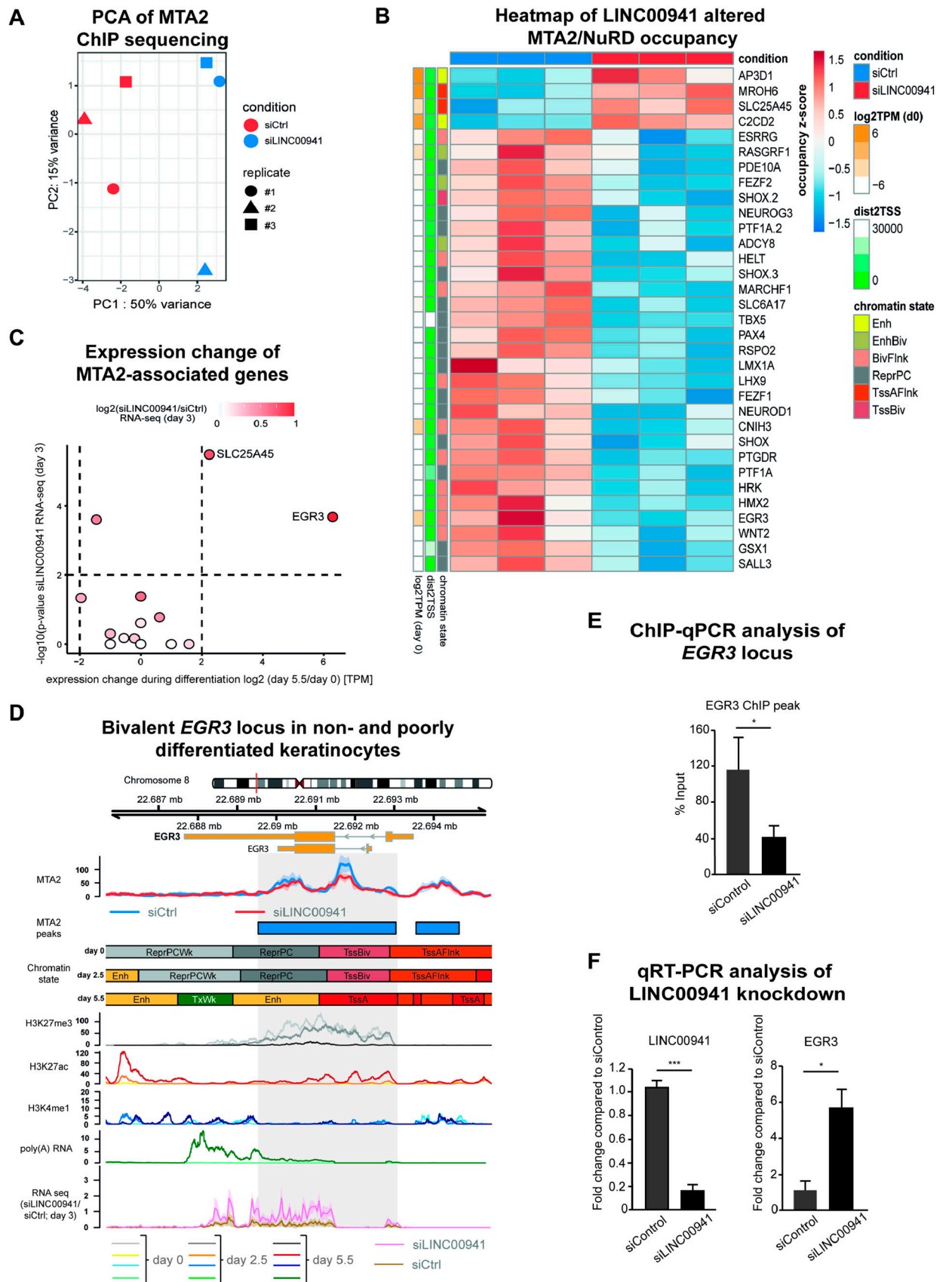

**A** PCA of MTA2 ChIP sequencing

**B** Heatmap of LINC00941 altered MTA2/NuRD occupancy

**C** Expression change of MTA2-associated genes

**D** Bivalent *EGR3* locus in non- and poorly differentiated keratinocytes

**E** ChIP-qPCR analysis of *EGR3* locus

**F** qRT-PCR analysis of LINC00941 knockdown

sites, thus allowing for fine-tuned regulation of biological processes, such as epidermal differentiation. We could show epigenetic repression of *EGR3* transcription in undifferentiated progenitor keratinocytes because of MTA2/NuRD-mediated gene silencing, with MTA2/NuRD occupancy being dependent on LINC00941. However, the absence of LINC00941-mediated MTA2/NuRD binding to the *EGR3* gene locus facilitated its expression and resulted in increased abundance of its downstream targets involved in keratinocyte differentiation (Figs 4D and F, 5, and 6).

Similar to the above study, LINC00941 was previously found to also influence mRNA transcription. However, this preceding study described the involvement of LINC00941 in *cis*-acting processes: LINC00941 regulates the expression of its nearby gene *CAPRIN2* through interacting with CTCF, a transcriptional regulator mediating chromatin looping (Ai et al, 2020). Another work revealing a role of LINC00941 as a regulator of mRNA transcription was recently published: Lu et al (2023) described LINC00941-mediated recruiting of ILF2 and YBX1 to the *SOX2* promoter region to enhance its transcription. In both cases, LINC00941 was found to have an activating gene regulatory role in both oral squamous and esophageal cell carcinoma, respectively. Here, the interaction of LINC00941 with NuRD-associated MTA2, as well as CHD4, could be demonstrated. Predominantly acting as a transcriptional repressor in non-cancerous epithelial cells and tissues bound to the NuRD complex, LINC00941 is thus another lncRNA that plays an important role in epidermal homeostasis, in addition to the lncRNAs ANCR, TINCR, SMRT-2, BLNCR, and PRANCR (Kretz et al, 2012, 2013; Lee et al, 2018; Tanis et al, 2019; Cai et al, 2020).

We previously showed that LINC00941-regulated genes associated with the process of epidermal stratification and differentiation—including genes located within the epidermal differentiation cluster (Ziegler et al, 2019). Interestingly, our results presented above did not reveal direct occupancy of the NuRD complex at genes within the EDC. Instead, we found LINC00941-controlled MTA2/NuRD occupancy at gene loci of various transcription factors, including *EGR3*. Correspondingly, LINC00941 likely acts through epigenetic repression of EGR3, a transcriptional regulator of differentiation gene expression. Our data indicate that

*EGR3* transcription was likely repressed in non- and poorly differentiated strata of human epidermis through binding of LINC00941/MTA2. Because MTA2 and CHD4 — both are core components of the NuRD complex — as well as LINC00941, abundance decreased upon differentiation (Fig 2A), epigenetic repression of *EGR3* transcription was reduced (Fig 4D). This likely resulted in EGR3-mediated transcription of downstream targets, including early and late differentiation genes—many of which are located within the EDC and induced in LINC00941-deficient epidermis — as demonstrated by the analyses presented here (Fig 6) and others (Kim et al, 2019).

As part of the above study, we found enrichment of LINC00941/MTA2/NuRD occupancy to bivalent and repressed chromatin states, respectively, including the *EGR3* gene locus (Fig 4B). Bivalent chromatin states were first described by Bernstein et al (2006) as they discovered nucleosomes marked with both repressing H3K27me3 and activating H3K4me3 histone modifications at the same time. In mouse embryonic stem (ES) cells, they found these bivalent domains frequently in close proximity to non- and poorly expressed developmental transcription factor genes and other genes important for development, which were poised for subsequent activation during ES cell differentiation. However, these repressed regions resolved during ES cell differentiation (Bernstein et al, 2006). Therefore, it was initially proposed that bivalent histone modifications keep developmental genes in a silenced but poised state, thus allowing either rapid activation or stable silencing during ES cell differentiation. In the later course, this definition was revised because bivalent domains were discovered not only in mouse and human ES cells but also in mammalian adult stem cells and adult tissues including keratinocytes (Mikkelsen et al, 2007; Barrero et al, 2013; Kinkley et al, 2016). Consequently, bivalency is now commonly seen as a way of expression fine-tuning during cell development and cell fate decisions, respectively, to prevent unscheduled gene activation providing robustness and plasticity but reduced noise during repression (Voigt et al, 2013). One of the central components in establishing and maintaining bivalency is the PcG proteins PRC1 and PRC2 (Harikumar & Meshorer, 2015), which are thought to be recruited by lncRNAs, among others (Voigt

---

**Figure 4. LINC00941 dependency of NuRD-associated MTA2 binding in *EGR3*.**
**(A)** Principal component analysis of MTA2 ChIP-sequencing experiments upon LINC00941 knockdown. **(B)** Heatmap illustrating differential MTA2-bound sites (*n* = 33 at an FDR of 5%) upon LINC00941 knockdown. MTA2/NuRD showed different chromatin occupancy in LINC00941-deficient cells compared with control. Z-scores of normalized fragment counts under differential MTA2 binding sites are indicated by the color code. Row labels on the right depict the gene nearest to the differential MTA2 binding site. Associated chromatin state (chromHMM) and distance to TSS (dist2TSS) or expression level (log$_2$[TPM] at day 0) in primary keratinocytes of associated genes are indicated on the left side. Abbreviations of chromatin states: Tss, transcription start site; A, active; Flnk, flanking; Biv, bivalent; Enh, enhancer; Repr, repressed; PC, Polycomb. **(C)** Scatter plot illustrating expression change in keratinocyte differentiation or upon LIN00941 knockdown of MTA2-associated genes. EGR3 showed the strongest changes upon both keratinocyte differentiation and LINC00941 knockdown. The x-axis represents the log$_2$ fold change in expression after day 5.5 of calcium-induced keratinocyte differentiation against primary keratinocytes. TPM values were obtained from ENCODE portal (day 0: ENCFF423MWU; day 5.5: ENCFF379PNP). The y-axis denotes the –log$_{10}$ transformed *P*-values derived from differential gene expression analysis between cells treated with siLINC00941 and control keratinocytes after 3 d of differentiation induction (Ziegler et al, 2019). The log$_2$ fold change in expression upon LINC00941 knockdown is indicated by shades of red. Genes exhibiting a *P*-value < 0.01 and |log$_2$ fold change| > 2 are highlighted. **(D)** Genome browser view of differential MTA2 binding sites at the *EGR3* locus. *EGR3* showed bivalent chromatin states in non- and poorly differentiated keratinocytes. Tracks of chromatin states, histone modifications, and transcription in primary undifferentiated (day 0) and calcium-treated differentiated (day 2.5, day 5.5) keratinocytes obtained from ENCODE portal are shown below the MTA2 ChIP-sequencing tracks. Color shades in H3K27me3, H3K27ac, H3K4me1, and RNA-sequencing tracks indicate the time points of differentiation: light shade for day 0, medium shade for day 2.5, and dark shade for day 5.5. The bottom track shows the RNA-sequencing coverage of siLINC00941-transfected (pink) and control (brown) keratinocytes after day 3 of differentiation. The dashed box highlights the differential MTA2 binding site at *EGR3*. Abbreviations: Tss, transcription start site; A, active; Flnk, flanking; Biv, bivalent; Tx, transcription; Wk, weak; Enh, enhancer; Repr, repressed; PC, Polycomb; Quies, quiescent. **(E)** ChIP-qPCR data analysis of the *EGR3* gene locus showed decreased MTA2 chromatin occupancy in siLINC00941 cells compared with control-treated cells. ChIP-qPCR data were normalized relative to input samples (*adj. *P* < 0.05). **(F)** siPool-mediated knockdown of LINC00941 resulted in a decreased *EGR3* level in organotypic epidermal tissues (*n* = 3–5 tissue cultures/knockdown group). Data are presented as the mean ± SD. Statistical significance was tested by an unpaired *t* test and corrected for multiple testing after Bonferroni (***adj. *P* < 0.001 and *adj. *P* < 0.05).

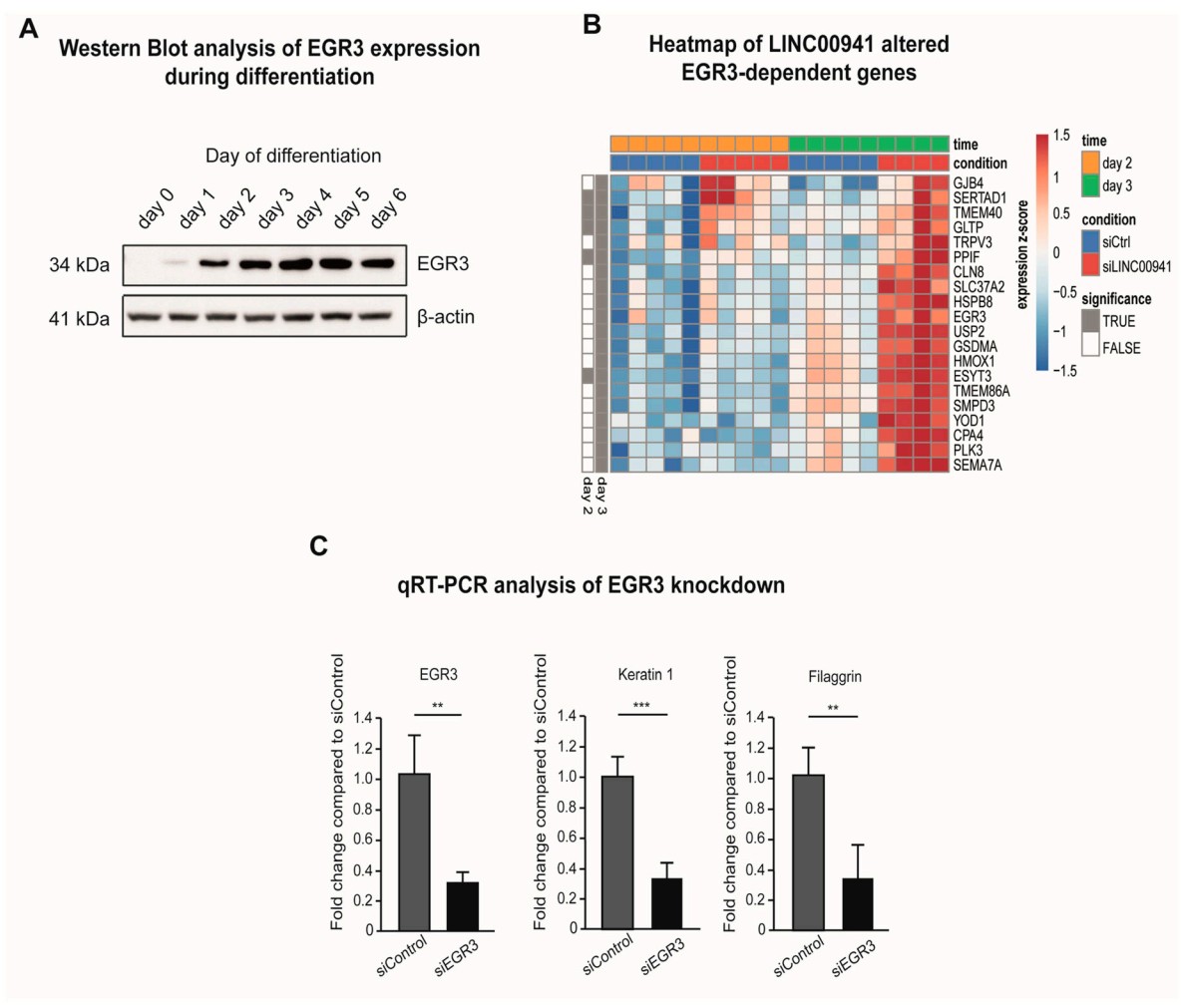

**A** Western Blot analysis of EGR3 expression during differentiation

Day of differentiation

day 0  day 1  day 2  day 3  day 4  day 5  day 6

34 kDa — EGR3

41 kDa — β-actin

**B** Heatmap of LINC00941 altered EGR3-dependent genes

GJB4
SERTAD1
TMEM40
GLTP
TRPV3
PPIF
CLN8
SLC37A2
HSPB8
EGR3
USP2
GSDMA
HMOX1
ESYT3
TMEM86A
SMPD3
YOD1
CPA4
PLK3
SEMA7A

expression z-score

time
day 2
day 3

condition
siCtrl
siLINC00941

significance
TRUE
FALSE

day 2
day 3

**C** qRT-PCR analysis of EGR3 knockdown

EGR3 — Fold change compared to siControl — ** — siControl, siEGR3

Keratin 1 — Fold change compared to siControl — *** — siControl, siEGR3

Filaggrin — Fold change compared to siControl — ** — siControl, siEGR3

**D** Immunofluorescence analysis of organotypic epidermal tissue

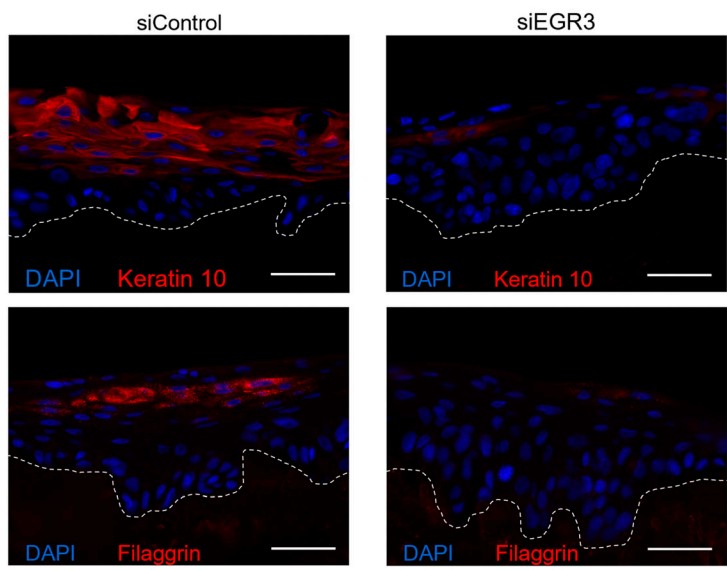

siControl    siEGR3

DAPI  Keratin 10    DAPI  Keratin 10

DAPI  Filaggrin    DAPI  Filaggrin

et al, 2013). In case of the NuRD complex, its role in interrelation with PcG proteins is better explored. It could be demonstrated that this chromatin remodeler deacetylates H3K27 in ES cells enabling PcG protein recruitment. This, in turn, can catalyze trimethylation of H3K27 at NuRD target genes allowing cell differentiation (Kaji et al, 2006; Hu & Wade, 2012; Reynolds et al, 2012; Kim et al, 2015). Consequently, the NuRD complex controls the sensitive equilibrium between acetylation and methylation of histones, thus influencing the expression of genes critical for cell fate decisions. These findings are in accordance with our results because we also detected MTA2/NuRD at active, repressed, and bivalent chromatin states (Fig 3C). For the first time, however, we could show that the MTA2/NuRD binding at bivalent domains in close proximity to transcriptional regulators, such as the transcription factor gene *EGR3*, depends on the presence of a lncRNA. Therefore, we hypothesize that LINC00941 might directly influence the maintenance of bivalent regions as it recruits NuRD. The resulting repression of *EGR3* is normally maintained in non- and poorly differentiated keratinocytes until this bivalent site is resolved through external factors of unknown origin. As LINC00941- and NuRD-associated MTA2 abundance decreased upon progressive differentiation, they were supposed to contribute to resolving bivalent domains. This assumption was further supported by the fact that LINC00941-deficient cells showed the premature expression of EGR3 coding for a transcription factor important for keratinocyte cell fate with regard to differentiation. This implied an early removal of the bivalent status. The precise mechanism of LINC00941-dependent NuRD complex recruitment and modulation of activity, respectively, is not known to date and represents a matter of ongoing research. However, recent studies have revealed CHD4 as an actual RNA-binding subunit of NuRD, thus tethering ncRNAs to target chromatin sites mediating DNA-RNA triplexes (Zhao et al, 2018; Ullah et al, 2022). In summary, our findings revealed the mode of action of LINC00941 involved in epidermal homeostasis. This process was dependent on LINC00941-mediated recruitment of NuRD-associated MTA2 to bivalent chromatin sites, which could also be found at the *EGR3* gene locus. LINC00941/MTA2-regulated repression of *EGR3* gene expression prevented premature keratinocyte differentiation.

# Materials and Methods

## Tissue culture

Pooled primary human keratinocytes from different juvenile donors were obtained from PromoCell. The tissue used by PromoCell for the isolation of human cells is derived from donors who have signed an informed consent form (by the donor, an authorized agent, or a legal agent) that outlines the purpose of the donation and the procedure for processing the tissue. PromoCell does not accept or use any tissue without prior signing of the consent document. The present analyses were conducted with pooled cell samples without personally identifiable information and deemed exempt from the requirement for IRB approval.

Keratinocytes were grown in a 1:1 mixture of KSF-M (Gibco) and Medium 154 for keratinocytes (Gibco). The media were supplemented with Human Keratinocyte Growth Supplement, bovine pituitary extract, human epidermal growth factor, and 0.5x antibiotic–antimycotic (all supplements were obtained from Gibco). Cells were cultured at 37°C in a humidified chamber with 5% $CO_2$. In vitro keratinocyte differentiation was induced by the addition of 1.2 mM calcium to the media and seeding cells at full confluency.

## RNA knockdown

siRNA pools of 11–30 different siRNAs were synthesized and obtained from siTools (Hannus et al, 2014). For siRNA-mediated knockdown, 5–6 million primary human keratinocytes were electroporated with 1 nmol siPools using the Amaxa human keratinocyte Nucleofector kit (Lonza) according to the manufacturer's instructions and the T-018 program of the Amaxa Nucleofector II device (Lonza). After nucleofection, cells recovered for 24 h.

## Organotypic human epidermal tissue

For the generation of organotypic human epidermal tissues, 550,000 human keratinocytes nucleofected with siRNAs were seeded onto a devitalized dermal matrix and raised to the air–liquid interface to initiate stratification and differentiation, as described previously (Truong et al, 2006; Sen et al, 2010).

## Immunofluorescence and tissue analysis

Seven-micrometer-thick paraffin-embedded cross sections of human organotypic skin cultures were deparaffinized and rehydrated by two changes of xylene and washed with descending ethanol concentrations. For antigen retrieval, the cross sections were boiled twice in demasking buffer (10 mM sodium citrate, pH 6.0, with 0.05% Tween-20) followed by blocking in PBS with 10% BCS for 20 min at RT. Antibodies were diluted in PBS with 1% BCS and incubated with the sections for 1 h. Primary antibodies included the

**Figure 5. EGR3 induces differentiation of human organotypic epidermis.**
**(A)** Western blot analysis of EGR3 repression during calcium-induced differentiation in primary keratinocytes (days 1–6) compared with undifferentiated keratinocytes (day 0). **(B)** Heatmap showing transcriptional changes in EGR3 and EGR3-regulated genes in keratinocyte differentiation upon LINC00941 knockdown. Genes regulated by EGR3 in keratinocytes were selected based on the study of Kim et al (2019). Expression data of siLINC00941-treated keratinocytes after days 2 and 3 of differentiation were obtained from GEO (GSE118077) (Ziegler et al, 2019) and reanalyzed. Significant expression changes (an FDR of 5%) of pairwise comparisons between siLINC00941 and control at each time point are indicated at the left sidebar (gray = significant, white = not significant). **(C)** qRT–PCR analysis of EGR3 knockdown. siPool-mediated knockdown of EGR3 in primary keratinocytes resulted in decreased mRNA abundance of early and late differentiation markers keratin 1 and filaggrin on day 3 of differentiation in organotypic epidermal tissues (*n* = 4–6 tissue cultures/knockdown group). Data are presented as the mean ± SD. Statistical significance was tested by an unpaired *t* test and corrected for multiple testing after Bonferroni (***adj. *P* < 0.001 and **adj. *P* < 0.01). **(D)** Immunofluorescence (IF) analysis showed a decreased level of early and late differentiation markers keratin 10 and filaggrin. The dashed line indicates the basement membrane, nuclei are shown in blue, and the differentiation proteins keratin 10 and filaggrin are shown in red (*n* = 4–6 tissue cultures/knockdown group, day 3 of differentiation, one exemplary picture for each group is depicted). Scale bar: 100 $\mu$m.

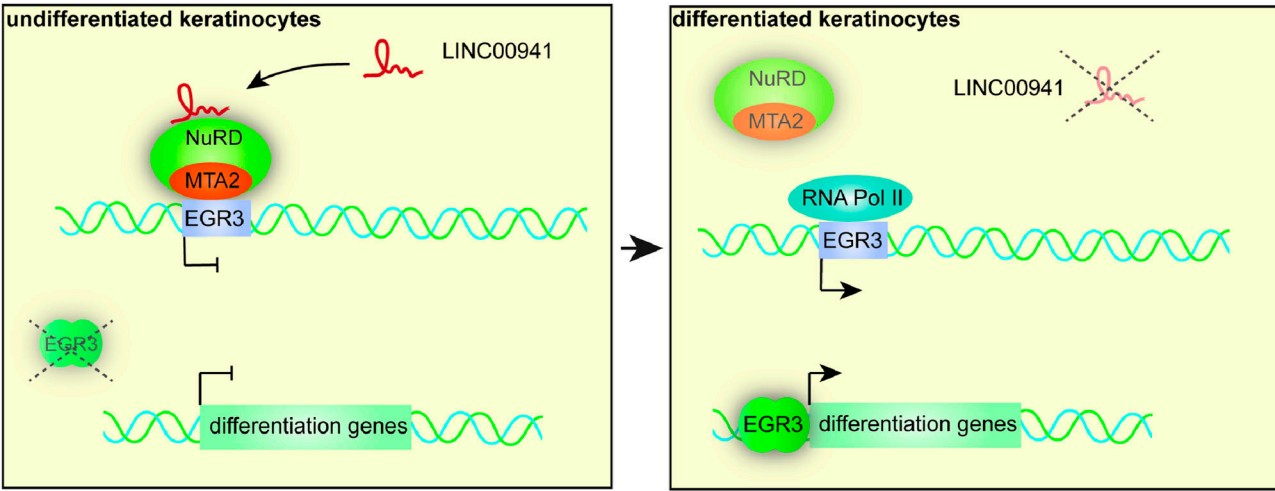

**Figure 6. LINC00941 regulates epidermal differentiation genes through modulation of MTA2/NuRD at the *EGR3* gene locus.**
Predicted mechanism of LINC00941. The interaction of LINC00941 with the MTA2/NuRD complex modulates its binding to chromatin, thus repressing EGR3 expression in non- and poorly differentiated keratinocytes. During keratinocyte differentiation, LINC00941 and MTA2 abundance decreases and *EGR3* is actively transcribed, which in turn results in the expression of its downstream differentiation gene targets and consequentially acceleration of keratinocyte differentiation.

following: filaggrin (sc-66192; Santa Cruz) at 1:50 dilution, MTA2 (ab8106; Abcam) at 1:500 dilution, and keratin 10 (MS-611-P; Neo-Markers) at 1:400 dilution. Alexa Fluor 555–conjugated goat anti-mouse (A21422, 1:300; Life Technologies) was used as a secondary antibody diluted in PBS with 1% BCS. Unbound antibodies were washed off after 1 h of incubation with PBS (three times, 5 min, RT). Slides were mounted in DAPI Fluoromount-G (Biozol). Finally, cross sections were analyzed with a BZ-X800 fluorescence microscope and BZ-X800 Analyzer software (Keyence).

**qRT–PCR analysis**

Total RNA from organotypic skin cultures was isolated with the RNeasy Plus mini kit (QIAGEN) according to the manufacturer's instructions. Total RNA from keratinocytes was isolated with TRIzol (Invitrogen) as stated by the manufacturer. Quantified by Nano-Drop, 1 $\mu$g total RNA was subjected to reverse transcription with the iScript cDNA synthesis kit (Bio-Rad). In case of RNA isolated with TRIzol, genomic DNA was removed using DNase I (Thermo Fisher Scientific) before cDNA synthesis. For qRT–PCR measurements, the Takyon Mix (Eurogentec) was used with CFX96 Touch Real-time PCR Detection System (Bio-Rad). Samples were run at least in biological duplicates and normalized to ribosomal protein L32 mRNA according to the $\Delta\Delta C_t$ method (Livak & Schmittgen, 2001). The following primer sequences were used: CHD3_fwd: 5′-CCATCTCCTGAACTTCCTCACC-3′, CHD3_rev: 5′-CTTGAGTCTCCGCAG-CATGTGT-3′; CHD4_fwd: 5′-CTGTTGCTGACTGGGACACCAT-3′, CHD4_rev: 5′-TGGTCCTCCTTGGCAATGTCAG-3′; EGR3_fwd: 5′-ACGCCAAGATCCACCTCAAG-3′, EGR3_rev: 5′-GGAAAAGTGGGGATCTGGGG-3′; EGR3_ChIP_fwd: 5′-TCCCAAGTAGGTCACGGTCT-3′, EGR3_ChIP_rev: 5′-GGCTCAGCGTTCCTCT-TAC-3′; Filaggrin_fwd: 5′-AAAGAGCTGAAGGAACTTCTGG-3′, Filaggrin_rev: 5′-AACCATATCTGGGTCATCTGG-3′; GAPDH_fwd: 5′-GAAGAGAGA-GACCCTCACTGCTG-3′, GAPDH_rev: 5′-ACTGTGAGGAGGGGA-GATTCAGT-3′; Keratin1_fwd: 5′-TGAGCTGAATCGTGTGATCC-3′, Keratin1_rev: 5′-CCAGGTCATTCAGCTTGTTC-3′; L32_fwd: 5′-

AGGCATTGACAACAGGGTTC-3′, L32_rev: 5′-GTTGCACATCAGCAGCACTT-3′; LCE1A_fwd: 5′-GAAGCGGACTCTGCACCTAGAA-3′, LCE1A_rev: 5′-AGGAGA-CAGGTGAGGAGGAAATG-3′; LCE6A_fwd: 5′-GTCCTGATCTCTCCTCTCGTCT-3′, LCE6A_rev: 5′-CAAGATTGCTGCTTCTGCTGT-3′; LINC00941_fwd: 5′-GACCTTTTCAGGCCAGCATT-3′, LINC00941_rev: 5′-ACAATCTGGA-TAGAGGGCTCA-3′; MTA1_fwd: 5′-CCAGGACCAAACCGCAGTAACA-3′, MTA1_rev: 5′-GTCAGCTTCGTCGTGTGCAGAT-3′; MTA2_fwd: 5′-TGTACCGGGTGGGAGATTAC-3′, MTA2_rev: 5′-GCCTCCACATTTCCATTTGC-3′; MTA3_fwd: 5′-AGCCCACTTACGGATCGACAGA-3′, MTA3_rev: 5′-CAAACTAGGCTGCCTCACAGAAC-3′; SPRR4_fwd: 5′-AGCCTCCAA-GAGCAAACAGA-3′, SPRR4_rev: 5′-GCAGGAGGAGATGTGAGAGG-3′. Statistical significance was tested by an unpaired *t* test and corrected for multiple testing after Bonferroni (*adj. *P* < 0.05, **adj. *P* < 0.01, ***adj. *P* < 0.001, and ns = not significant).

**In vitro RNA pull-down with subsequent protein detection**

For in vitro RNA transcription of LINC00941, the Sp6 RNA polymerase was used according to the manufacturer's instructions with the exception that LINC00941 was labeled with biotin-16-UTP (Roche). 2 $\mu$g of biotin-16–labeled LINC00941 was incubated with a protein lysate of 3 × 10$^7$ keratinocytes. The protein lysate was prepared beforehand by collecting the cells in lysis buffer (50 mM Tris, pH 7.5, 150 mM NaCl, 5% glycine, 1 mM DTT, 1 mM EDTA, 0.1 U/$\mu$l RiboLock RNase Inhibitor [Thermo Fisher Scientific], 1 mM AEBSF, 1x cOmplete Protease Inhibitor Cocktail [Roche]) and clearing the lysates for 15 min, at full speed at 4°C. After 16 h at 4°C, 5 $\mu$l of MyOne Streptavidin C1 Dynabeads (Invitrogen) was added, and the mixture was incubated for 1 h at RT and subjected to three wash cycles of 10 min, each using 500 $\mu$l wash buffer (150 mM NaCl, 50 mM Tris, pH 7.5, 10% glycine, 5 $\mu$M EDTA, 0.5% Igepal, and 1x cOmplete Protease Inhibitor Cocktail [Roche]). After the final wash, magnetic beads were resuspended in 12 $\mu$l 1x Laemmli buffer with 10% 2-mercaptoethanol following the Western blot analysis.

## Western blot analysis

Protein lysates from calcium-induced differentiated keratinocytes were obtained by scraping cells into a suitable amount of lysis buffer (50 mM Tris, pH 7.5, 150 mM sodium chloride, 5% glycerin, 1 mM DTT, 1 mM EDTA, 0.1 U/$\mu$l RiboLock RNase Inhibitor [Thermo Fisher Scientific], 1 mM AEBSF, 1x cOmplete Protease Inhibitor Cocktail [Roche]). Lysates were cleared for 15 min, at full speed at 4°C, and the protein concentration was determined via the Bradford assay. 30 $\mu$g of total protein was separated on a 10% SDS–PAGE and transferred onto the Amersham Hybond ECL membrane (Cytiva Life Science) by semi-dry blot. Blocking and antibody dilution were done in 5% milk powder in TBS-T. Primary antibodies included MTA2 (ab8106; Abcam) at 1:1,000 dilution, EGR3 (ab232820; Abcam) at 1:500 dilution, CHD4 (ab72418; Abcam) at 1:5,000 dilution, and $\beta$-actin (ab6276; Abcam) at 1:10,000 dilution. Secondary antibodies used were IRDye 800 goat anti-rabbit (926-32211; LI-COR Biosciences) and IRDye 680 goat anti-mouse (926-32220; LI-COR Biosciences), both at a 1:15,000 dilution. Western blots were analyzed with Odyssey Infrared Imager (LI-COR Biosciences).

## Lentiviral-mediated LINC00941 overexpression

For HEK293T transfection, Lipofectamine 3000 (Thermo Fisher Scientific) was used according to the manufacturer's instruction. The lentiviral transfer vector pLARTA-LINC00941 and packaging construct vectors (pUG-MDG and pCMV-$\Delta$R8.91) were applied in equimolar quantities. Viral particles were harvested 48 and 72 h after transfection. For lentiviral transduction, 50,000 keratinocytes per well were seeded in a six-well plate the day before infection. The next day, an appropriate amount of viral particles diluted in keratinocyte medium and 5 $\mu$g/ml polybrene were added to the cells. Infection was accomplished in a centrifugation step at RT with 250 rcf for 1 h. Afterward, cells were recovered in kerati-nocyte medium for at least 24 h before their cell lysate was used for RNA-IP.

## MS analysis

MS analysis was essentially performed as previously described (Hoffmeister et al, 2023). Proteins were separated on a NuPAGE 4–12% Bis-Tris gel (Invitrogen) according to the manufacturer's instructions, and gel lanes were cut into consecutive slices. The gel slices were washed with 50 mM $NH_4HCO_3$, 50 mM $NH_4HCO_3$/ace-tonitrile (3/1), and 50 mM $NH_4HCO_3$/acetonitrile (1/1) while shaking gently in an orbital shaker (VXR basic Vibrax, IKA). Gel pieces were lyophilized after shrinking by 100% acetonitrile. To block cysteines, reduction with DTT was carried out for 30 min at 57°C followed by an alkylation step with iodoacetamide for 30 min at room temperature in the dark. Subsequently, gel slices were washed and lyophilized again as described above. Proteins were subjected to in-gel tryptic digest overnight at 37°C with ~2 $\mu$g trypsin per 100 $\mu$l gel volume (Trypsin Gold, MS grade; Promega). Peptides were eluted twice with 100 mM $NH_4HCO_3$ followed by an additional extraction with 50 mM $NH_4HCO_3$ in 50% acetonitrile. Before LC-MS/MS analysis, combined eluates were lyophilized and reconstituted in 20 $\mu$l of 1% formic acid. Separation of peptides by reversed-phase chromatography

was carried out on UltiMate 3000 RSLCnano System (Thermo Fisher Scientific), which was equipped with a C18 Acclaim PepMap 100 preconcentration column (100 $\mu$m i.D. × 20 mm, Thermo Fisher Scientific) in front of an Acclaim PepMap 100 C18 nano column (75 $\mu$m i.d. × 150 mm, Thermo Fisher Scientific). A linear gradient of 4–40% acetonitrile in 0.1% formic acid over 90 min was used to separate peptides at a flow rate of 300 nl/min. The LC system was coupled on-line to maXis plus UHR-QTOF System (Bruker Daltonics) via a CaptiveSpray nanoflow electrospray source (Bruker Daltonics). Data-dependent acquisition of MS/MS spectra by CID fragmenta-tion was performed at a resolution of minimum 60,000 for MS and MS/MS scans, respectively. The MS spectrum rate of the precursor scan was 2 Hz processing a mass range between m/z 175 and m/z 2,000. Using Compass 1.7 acquisition and processing software (Bruker Daltonics), a dynamic method with a fixed cycle time of 3 s and a m/z-dependent collision energy adjustment between 34 and 55 eV was applied. Raw data processing was performed in Data Analysis 4.2 (Bruker Daltonics), and Protein Scape 3.1.3 (Bruker Daltonics) in connection with Mascot 2.5.1 (Matrix Science) facili-tated database searching of the Swiss-Prot *Homo sapiens* database (release-2020_01, 220420 entries). Search parameters were as fol-lows: enzyme specificity trypsin with one missed cleavage allowed, precursor tolerance 0.02 D, MS/MS tolerance 0.04 D. Carbamido-methylation or propionamide modification of cysteine, oxidation of methionine, deamidation of asparagine and glutamine were set as variable modifications. Mascot peptide ion-score cutoff was set to 25. Search conditions were adjusted to provide an FDR of less than 1%. Protein list compilation was done using the Protein Extractor function of Protein Scape. EmPAI values (exponentially modified protein abundance index), which can be used for an approximate relative quantitation of proteins in a mixture, were extracted from Mascot. MS analysis resulted in $n$ = 627 LINC00941 interaction partners. GO term analysis was carried out using the PANTHER overrepresentation test (Thomas et al, 2022). GO terms with an FDR of less than 0.05 and chromatin association were considered for further analysis.

## RNA-IP

For RNA-IP, the cell lysate of in vitro differentiated keratinocytes overexpressing LINC00941 was obtained by scraping cells into a suitable amount of RIPA buffer. Lysates were cleared for 10 min, at full speed at 4°C. 5 $\mu$g of either $\alpha$-MTA2 antibody (ab8106; Abcam), $\alpha$-CHD4 (ab263025; Abcam), or IgG control (I5006; Santa Cruz) was added to the lysate and incubated at 4°C for 2 h. 40 $\mu$l Protein G Dynabeads (Invitrogen) were transferred to the lysate and incu-bated for 1 h at 4°C. After three washing steps with RIPA buffer, the beads were resuspended in TRIzol (Invitrogen), and RNA was isolated.

## Chromatin immunoprecipitation (ChIP) sequencing

For ChIP, primary keratinocytes were grown to a confluency of about 80% and dethatched from cell culture dishes. Following a cross-linking step adding 1% formaldehyde at RT for 10 min, the reaction was quenched with 125 mM glycine at RT for 5 min. Cells were pelleted and then lysed in swelling buffer (100 mM Tris, pH 7.5,

10 mM potassium acetate, 15 mM magnesium acetate, 1% Igepal, 1x cOmplete Protease Inhibitor Cocktail [Roche], and 1 mM AEBSF). Chromatin fragmentation was subsequently performed in RIPA buffer for 30 min using the S220 Focused-ultrasonicator (Covaris) in the Freq sweeping mode, intensity 8, 20% duty cycle, and 200 cycles per burst. 100 $\mu$g fragmented chromatin underwent IP with 5 $\mu$g $\alpha$-MTA2 antibody (ab8106; Abcam) for 16 h at 4°C. 40 $\mu$l Protein G Dynabeads (Invitrogen) were added to the IP and incubated for 45 min at RT. The beads were washed three times with wash buffer (100 mM Tris, pH 9.0, 0.5 M lithium chloride, 1% Igepal, 1% sodium deoxycholate, and 1 mM AEBSF) followed by the elution with elution buffer (50 mM sodium bicarbonate and 1% SDS). For reverse cross-linking, 0.2 M sodium chloride was added, and the chromatin was incubated at 67°C for 4 h following a purification step with the QIAquick PCR Purification kit (QIAGEN) according to the manufacturer's instructions. For library preparation, the NEBNext Ultra II DNA Library Preparation kit for Illumina (New England Biolabs) and NEBNext Multiplex Oligos for Illumina (New England Biolabs) were used following the manufacturer's instructions. The libraries were pooled equimolar, and the pool was quantified using the KAPA Library Quantification kit—Illumina (Roche). The libraries were sequenced on an Illumina NextSeq 2000 instrument (Illumina) controlled by NextSeq Control Software, v1.4.1.39716, using a 100 cycles P2 Flow Cell with the single-index, paired-end (PE) run parameters. Image analysis and base calling were done by Real Time Analysis Software, v3.9.25. The resulting .cbcl files were converted into .fastq files with bcl2fastq v2.20 software. ChIP sequencing was performed at the Genomics Core Facility "KFB–Center of Excellence for Fluorescent Bioanalytics" (University of Regensburg, Germany).

### ChIP-sequencing data analysis

Initially, quality control of the raw sequence reads was conducted using FastQC (v0.11.8) (Andrews, 2010). Subsequently, reads were mapped to the reference genome (GRCh38) using bowtie2 (v2.4.4). The following options were used to optimize the alignment process: --very-sensitive-local, --no-discordant, --no-mix, --dovetail. Aligned reads were filtered for MAPQ >= 30 using SAMtools (Li et al, 2009). Reads mapping to blacklisted genomic regions (ENCODE accession: ENCFF356LFX) were removed using BEDTools (Quinlan & Hall, 2010). Peak calling was executed on all samples relative to input samples using MACS2 software (Zhang et al, 2008). The specific options used for this analysis were as follows: –f BAMPE, –g 2.7e9, --keep-dup auto. After peak calling, the resultant peaks were filtered based on a log q-value threshold >20 for subsequent analyses. This stringent cutoff was used to ensure a high level of confidence in the identified peaks, minimizing the potential for false positives. The number of fragments in each sample falling under these peaks was quantified using the featureCounts function of the Subread package (Liao et al, 2014).

To identify changes in MTA2 binding upon LINC00941 knockdown, the generated count table was processed in R using the Bioconductor package DESeq2 (Love et al, 2014). 33 MTA2 binding sites were found to significantly change their binding abundance upon LINC00941 knockdown based on an FDR of 0.05.

### RNA-sequencing data analysis

Publicly available RNA-sequencing data of LINC00941 knockdown during keratinocyte differentiation (GSE118077) were reanalyzed using a Nextflow RNA-sequencing pipeline (Di Tommaso et al, 2017).

Initially, quality control of the raw sequence reads was conducted using FastQC (v0.11.8) (Andrews, 2010). Subsequently, reads were mapped to the reference genome (GRCh38) and the corresponding gene annotation (Ensembl version 106) using Spliced Transcripts Alignment to a Reference software (v2.7.8a) (Dobin et al, 2013). The following options were used to optimize the alignment process: --outFilterType BySJout, --outFilterMultimapNmax 20, --alignSJoverhangMin 8, --alignSJDBoverhangMin 1, --outFilterMismatchNmax 999, --alignIntronMin 10, --alignIntronMax 1,000,000, --outFilterMismatchNoverReadLmax 0.04, --runThreadN 12, --outSAMtype BAM SortedByCoordinate, --outSAMmultNmax 1, and --outMultimapperOrder Random.

Post-mapping quality control was performed using the rnaseq mode of Qualimap (v2.2.1) (García-Alcalde et al, 2012). The level of PCR duplication was assessed using Picard MarkDuplicates (v2.21.8) (Broad Institute, 2019) and dupRadar (v1.15.0) (Sayols et al, 2016). Gene expression quantification was carried out using featureCounts (v1.6.3) (Liao et al, 2014).

Finally, differential gene expression analysis between LINC00941 knockdown and control samples was conducted using the Bioconductor DESeq2 package (Love et al, 2014). The normal shrinkage method was applied for scaling $\log_2$ fold changes (Zhu et al, 2019). Genes with an FDR of less than 0.05 were considered significantly differentially expressed. This analysis resulted in the identification of 385 down-regulated and 1,532 up-regulated genes in the LINC00941 knockdown keratinocytes at day 3.

### Downstream analysis

ChIP-sequencing coverage tracks were generated using the bamCoverage function from the deepTools package (Ramírez et al, 2016). For normalization, the size factors derived from DESeq2 analysis were used. Data track visualizations were generated using the Bioconductor package Gviz (Hahne & Ivanek, 2016). The Bioconductor package ChIPseeker was used to annotate the genomic regions of the MTA2 binding sites (Yu et al, 2015). To link the MTA2 binding sites to the next gene, only protein-coding genes were considered. Gene set overrepresentation analysis was carried out on MTA-associated genes with a maximum distance to the TSS of 3 kb using the Bioconductor package clusterProfiler (Wu et al, 2021).

### External datasets used in this study

The 15-state core chromatin model of male keratinocytes (roadmap accession: E057) was obtained from the roadmap repository (Roadmap Epigenomics Consortium et al, 2015). The corresponding histone modification and RNA-sequencing data tracks were obtained from ENCODE repository (The ENCODE Project consortium, 2012; Luo et al, 2020): poly(A) RNA-sequencing plus/minus strand (ENCFF283IQC/

ENCFF804BRH), H3K4me3 (ENCFF517FHJ), H3K27me3 (ENCFF400FLX), H3K4me1 (ENCFF319BIJ).

Data on in vitro differentiation of foreskin keratinocytes (at day 0/day 2.5/day 5.5) were obtained from the ENCODE portal: ChromHMM 18-state model (ENCFF571LVQ/ENCFF385FJV/ENCFF058GJN), H3K4me1 (ENCFF539WZS/ENCFF546FWF/ENCFF821VJZ), H3K27ac (ENCFF786TBH/ ENCFF870GTM/ENCFF988UOV), H3K27me3(ENCFF492WYD/ENCFF846IJG/ ENCFF902OOG), poly(A) RNA-sequencing plus (ENCFF646IJP/ ENCFF786AVM/ENCFF533VFT), poly(A) RNA-sequencing minus (ENCFF699EVT/ENCFF701XBY/ENCFF366JLV), poly(A) RNA-sequencing quantification (ENCFF423MWU/ENCFF137YHI/ENCFF379PNP).

RNA-sequencing data of LINC00941 knockdown after days 2 and 3 were obtained from Gene Expression Omnibus (GEO; GSE118077) (Ziegler et al, 2019).

## Data Availability

The datasets and computer code produced in this study are available in the following databases: MS data have been deposited to the ProteomeXchange Consortium via the PRIDE repository and assigned the accession number PXD043838. The code used to analyze the sequencing data is available at GitHub (https:// github.com/uschwartz/linc00941_MTA2_kerationcytes). Raw and processed ChIP-sequencing data have been deposited to the NCBI Gene Expression Omnibus (GEO) database and assigned the identifier GSE237175.

## Supplementary Information

## Acknowledgements

We thank Gernot Längst for critically reading this article. Our research is supported by the Deutsche Forschungsgemeinschaft (SFB 960, project B09 to M Kretz). We further thank the ENCODE Consortium and the Manolis Kellis and Michael Snyder Lab for producing the datasets of human keratinocytes used in this study. We thank Petra Richter for technical assistance.

### Author Contributions

E Morgenstern: conceptualization, data curation, formal analysis, validation, investigation, visualization, methodology, and writing—original draft.
C Molthof: data curation, formal analysis, validation, investigation, visualization, and methodology.
U Schwartz: software, formal analysis, and visualization.
J Graf: conceptualization, data curation, investigation, and methodology.
A Bruckmann: data curation and investigation.
S Hombach: conceptualization, validation, methodology, and writing—review and editing.
M Kretz: conceptualization, data curation, formal analysis, supervision, funding acquisition, investigation, visualization, and writing—original draft and project administration.

## Conflict of Interest Statement

The authors declare that they have no conflict of interest.

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
