## [Reviewer comments · Life Science Alliance]

Life Science Alliance

LncRNA LINC00941 modulates MTA2/NuRD occupancy to suppress premature human epidermal differentiation

Eva Morgenstern, Carolin Molthof, Uwe Schwartz, Johannes Graf, Astrid Bruckmann, Sonja Hombach, and Markus Kretz
DOI: <https://doi.org/10.26508/lsa.202302475>

Corresponding author(s): Markus Kretz, Medical School Hamburg

Review Timeline:

Submission Date:	2023-11-08
Editorial Decision:	2023-11-09
Revision Received:	2024-03-07
Editorial Decision:	2024-04-03
Revision Received:	2024-04-09
Accepted:	2024-04-10

Transaction Report:

Please note that the manuscript was reviewed at *Review Commons* and these reports were taken into account in the decision-making process at *Life Science Alliance*.

Reviews

Review #1

In this manuscript, Morgenstern et al investigated the molecular mechanism by which LINC00941 regulates keratinocyte differentiation. They found the LINC00941 interacts with the NuRD chromatin remodeling complex in human primary keratinocytes. Furthermore, LINC00941 silencing by RNAi results in changes in the genomic occupancy of MTA2, a core NuRD subunit, especially near a number of bivalent genes. In particular, they showed that LINC00941 depletion resulted in reduced MTA2 occupancy at the EGR3 gene, increased EGR3 expression, and increased expression of EGR3-regulated epidermal differentiation genes. Together, they propose that LINC00941 prevents premature differentiation of human epidermal tissue by repressing EGR3 expression in non-differentiated keratinocytes via NuRD. The interaction between LINC00941 and NuRD is a novel finding and will likely provide new insights for the function of LINC00941, which has been implicated in keratinocytes, tissue homeostasis and cancer. It will also shed light on the role of lncRNAs in epigenetic gene regulation and cell fate transition in general. The conclusion of this study can be much strengthened if the authors can identify LINC00941-occupied genomic regions by ChIRP (PMID: 21963238) or RAP (PMID: 23828888). In addition, the authors are also encouraged to address the following questions and comments to further improve the manuscript.

Fig 1C: Since multiple NuRD subunits were identified in the LINC00941 pull-down (Fig 1B), can the authors validate at least one other subunit? CHD4 is also a NuRD-specific subunit and appears to be a strong hit based on supplemental Fig 1.

Fig 1D: Can the authors also try RNA-IP on MTA2 and endogenous LINC00941?

Fig 2B: It seems that MTA2 protein level still remains reasonably high at day-4.

Fig 3C: How many bivalent promoters are there in keratinocytes? How many of those are bound by MTA2?

Fig S3A: Can the authors examine MTA2 occupancy at TSS and bivalent TSS in control vs. siLINC00941 cells (by meta-gene analysis)? This will show whether LINC00941 KD affects MTA2 occupancy at bivalent TSS in general.

Fig 4B: Does LINC00941 KD only affect 33 out of the 3613 MTA2 peaks? If yes, can the authors comment on why only such a small fraction of MTA2 occupied regions are affected?

Fig 4C: The authors only examined a small number of MTA2-associated genes. To provide a more complete view of the potential involvement of LINC00941-regulated genes in keratinocytes differentiation, can the authors provide the total number of differentially expressed genes (DEGs) in LINC00941 KD, the total number of DEGs during keratinocytes differentiation, and the overlap between the two (maybe using a venn diagram)? In addition, among all the overlapping DEGs from above, how many of them have MTA2 peaks nearby? Finally, in the overlapping DEGs occupied by MTA2, can the authors compare MTA2 occupancy at up- vs. down-regulated DEGs caused by LINC00941 KD, to see whether reduced MTA2 occupancy associates with increased expression after LINC00941 KD?

Fig 4D: Can the authors add the H3K4me3 track to the figure? Can the authors provide ChIP-qPCR result to validate the changes in MTA2 occupancy near EGR3 after LINC00941 KD?

Fig 5A: Some of the EGR3 target genes (eg., GJB4, SERTAD1) appear to be expressed before EGR3 up-regulation in siCtrl, and some of them (eg. HMOX1, ESYT3, SMPD3) appear to show stronger up-regulation than EGR3 in siLINC00941. This is not entirely consistent with the idea that they are regulated by LINC00941 via EGR3.

The interaction between LINC00941 and NuRD is a novel finding and will likely provide new insights for the function of LINC00941, which has been implicated in keratinocytes, tissue homeostasis and cancer. It will also shed light on the role of lncRNAs in epigenetic gene regulation and cell fate transition in general. The conclusion of this study can be much strengthened if the authors can identify LINC00941-occupied genomic regions by ChIRP (PMID: 21963238) or RAP (PMID: 23828888). In addition, the authors are also encouraged to address the above-mentioned questions and comments to further improve the manuscript.

Review #2

In this manuscript, the authors investigate the role of LINC00941 epidermal differentiation. Specifically the authors show interaction with MTA2 and other NuRD subunits. Next, the authors show that LINC00941 and MTA2 restricts premature keratinocyte differentiation, where KD of either results in increased differentiation marker expression. To understand

molecular impacts, the authors perform ChIPseq of MTA2 in control and LINC00941 depletion. Curiously, MTA2 binds in a trend differing from other cell types with predominant binding over active promoters. Upon LINC00941 KD, MTA2 binding is changed at 33 locations, where the majority show reduced binding. Overlapping binding changes with gene expression changes, the authors identify EGR3 as the only direct candidate upregulated upon LINC00941 KD and upregulated during differentiation. KD of EGR3 results in opposite trends of LINC00941 KD, suggesting the proposed mechanism of LINC00941 repressing EGR3 until appropriate time in differentiation. I have the following suggestions for this work:

1. While data support MTA2 acting in NuRD, beyond Fig 1, the authors exclusively use MTA2 as a proxy for NuRD. Of course there are some subunits that are within other complexes and should not be used, others are options. While I do not expect the authors to perform all experiments on an additional subunit of NuRD, I do think there are a few things the authors should consider:
 - a. Be more precise with language to point out only MTA2 rather than say NuRD complex throughout many aspects of the paper, and only assume the complex in limited settings and when it is clear it is speculative
 - b. Perform a subset of experiments on another subunit. For example, the Mass Spec in Fig 1A/B shows an interaction with other subunits, but the verification was only done for MTA2 (Fig 1C/D). This could easily be blotted (or another Western performed) and/or primers for other subunits for the qPCR for a couple additional subunits. Similarly straightforward, looking at MTA2 RNA expression changes during differentiation (Fig 2A): if additional primers were used to other subunits, these additional subunits could be used to verify.
2. Related to the above comment, does MTA2 KD (or LINC00941 KD for that matter) result in loss of NuRD complex formation? If so, this would be sufficient to address point 1.
3. Finally, in relation to NuRD complex here, it is important to note that mutually exclusive NuRD complexes (MBD2/NuRD and MBD3/NuRD) have been documented. Because the Mass Spec did not show interaction to MBD2 or MBD3, it is not clear if this is limited to one of these complexes. Related to this, the authors show by Mass spec that LINC00941 interacts with CHD4, but not CHD3. Is this because Chd3 is not expressed in these cells, or because there is some mutual exclusivity to CHD4 and LINC00941 is acting through this subcomplex?
4. Immunofluorescence images showing increased Keratin 10 and Filaggrin in LINC00941 or MTA2 KD (Fig 2E) and decreased Keratin 10 and Filaggrin in EGR3 KD (Fig 5C) are curious as the control look very different. In 2E, the control shows barely detectable levels, whereas in 5C the levels look similar to what is seen in Fig 2E KDs. Is this variability? If so, more representative images as well as quantification to the changes are necessary to make these two points.
5. In Figure 4, the authors present ChIPseq data for MTA2 in LINC00941 KD. One interesting trend is that the KD alters binding of MTA2 at mostly bivalent/repressed locations, rather than at active locations which is the majority of MTA2 binding (from Fig 3). It would be nice to show then these data rather than only stating it. The authors include a browser track for 2 genes (Fig 4D and S4C), but for the other 31 locations, a heatmap or something to show the level of K27me3 vs K27ac/K4me3 would be helpful to support this claim. Notably saying "Most of the differential MTA2/NuRD occupied sites were marked by repressive histone modification H3K27me3..." is the point that doesn't seem to be shown, and also a precise number should be included.
 - a. Related to this, I believe the authors performed K27me3 ChIPseq in the KD, and if so, it would be nice to see more genome wide effects here.
6. This is perhaps beyond the scope of the paper, but the obvious question to me is if EGR3 is relocalized in LINC00941 KD. Specifically, we would anticipate that EGR3 localization in the KD would mimic that of a more differentiated cell (binding to differentiated genes). A quick ChIPqPCR experiment for a few locations would be sufficient to support this model.

****Minor points:****

1. Importantly, CHD5 can also be incorporated in NuRD, in place of CHD3 or CHD4.
2. The authors use heatmaps and metaplots in Fig S3 to show reproducibility of the ChIPseq datasets. Importantly, the PCA does show some variation. XY scatterplots for replicates vs one another would be a more robust QC.
3. In figure 4D, the authors present nice data showing changes in histone mods during differentiation, but it is very hard to see the color changes and the tracks as presented. (same point for Fig S4C)
4. it is unclear from the methods or the figure legend if RTqPCR data are biological or technical replicates.

In this manuscript, the authors present a molecular function for LINC00941 in epidermal differentiation, where it interacts directly with NuRD subunit MTA2. LINC00941 has been previously described but this activity was not described. LINC00941 seems to specifically help target or maintain MTA2 localization to EGR3 to promote repression of this gene. Then, the model suggests that during differentiation, LINC00941 and MTA2 levels decrease, permitting activation of EGR3 during epidermal differentiation and subsequent activation of appropriate genes. These findings will be of interest to individuals interested in NuRD function, lncRNA activity and/or epidermal cell fate.

Review #3

In the manuscript entitled "The long non-coding RNA LINC00941 modulates MTA2/NuRD occupancy to suppress premature human epidermal differentiation", Morgenstern and colleagues describe a novel mechanism by which Nur complex interacting with a ncRNA LINC00947 represses expression of a late differentiation transcription factor EGR3 in the skin. These findings are novel and will be of interest in the field. Of note, the authors use a plethora of biochemical and cell biology techniques such as chromatin occupancy, transcriptomics and organotypic skin culture. Both the Introduction and Discussion outline a necessary background in the field of ncRNAs and skin biology as well as clearly state and discuss the scientific problem. The Methods are comprehensive and accurate. Importantly, any possible ethical issues are discussed. The Results are clear and support the data shown. Also the Figures are well organised and easy to follow. Overall, the manuscript is well prepared and will be of interest for broad readership. Before publication, however, the authors should address some points to further enhance their work:

****Major points:****

1. It is peculiar that there is no evident K10 and FLG staining in the organotypic skin from siControl cells in Fig 2E but it is present on the Fig 5C. The authors should explain these inconsistencies. Ideally the authors should provide a western blot analysis of these proteins which would help to give a more quantitative picture of the phenomenon.
2. There are no evidences at protein level of an efficiency of MTA2 and EGR3 knock-down. The authors use an MTA2 for western blot, so it should not be difficult to confirm. For EGR3, it is essential, as EGR3 is expected to increase only during late differentiation, while experiments from Fig 5B are performed only at 3 days of differentiation. Is it enough to induce EGR3 expression? Is the knock-down efficient at protein level at that time point? This is important as the work by Kim et al 2019 shows a detectable expression of EGR3 only after 7 days of differentiation.
3. The authors should demonstrate an increase of EGR3 at protein level after LINC00947 knock-down (by western blot in vitro and/or in epidermal organotypic tissue).
4. A key experiment to confirm the proposed model should be a double knock-down of both LINC00941 and EGR3. Will it rescue the observed increase of pro-differentiation genes?

****Minor points:****

1. 4E - EGR3 or LINC00941 knock-down?
2. 4E - The RNA seq label states "RNA seq (siLINC00941 d3)" but apparently shows both scr and siLINC00941
3. Please add the recipe of the lysis buffer used for western blot analysis of keratinocytes lysates.
4. Page 10 - Fig 4D description - is it "chromatin conformation state" or just "chromatin state"?

November 9, 2023

Re: Life Science Alliance manuscript #LSA-2023-02475

Dr. Markus Kretz
University of Regensburg; MSH Medical School Hamburg
Hamburg
Germany

Dear Dr. Kretz,

Thank you for submitting your manuscript entitled "The long non-coding RNA LINC00941 modulates MTA2/NuRD occupancy to suppress premature human epidermal differentiation" to Life Science Alliance. We would like to invite further consideration of this manuscript at LSA, revised according to your Revision Plan.

Thank you for this interesting contribution to Life Science Alliance. We are looking forward to receiving your revised manuscript.

Sincerely,

B. MANUSCRIPT ORGANIZATION AND FORMATTING:

Response to reviewer comments

We kindly thank all reviewers for their positive and helpful responses to this manuscript. Furthermore, we appreciate the concurring comments of the reviewers regarding the novelty and significance of our work describing a molecular mechanism of lncRNA LINC00941. Through interaction with NuRD-associated MTA2 in human keratinocytes, this lncRNA controls epigenetic gene regulation which subsequently affects epidermal homeostasis.

Below, we have included point-by-point responses to all comments of the three reviewers.

Changes in the text of the revised manuscript are highlighted in yellow.

Reviewer #1 (Evidence, reproducibility and clarity (Required)):

In this manuscript, Morgenstern et al investigated the molecular mechanism by which LINC00941 regulates keratinocyte differentiation. They found the LINC00941 interacts with the NuRD chromatin remodeling complex in human primary keratinocytes. Furthermore, LINC00941 silencing by RNAi results in changes in the genomic occupancy of MTA2, a core NuRD subunit, especially near a number of bivalent genes. In particular, they showed that LINC00941 depletion resulted in reduced MTA2 occupancy at the EGR3 gene, increased EGR3 expression, and increased expression of EGR3-regulated epidermal differentiation genes. Together, they propose that LINC00941 prevents premature differentiation of human epidermal tissue by repressing EGR3 expression in non-differentiated keratinocytes via NuRD. The interaction between LINC00941 and NuRD is a novel finding and will likely provide new insights for the function of LINC00941, which has been implicated in keratinocytes, tissue homeostasis and cancer. It will also shed light on the role of lncRNAs in epigenetic gene regulation and cell fate transition in general.

Comment1: The conclusion of this study can be much strengthened if the authors can identify LINC00941-occupied genomic regions by ChIRP (PMID: 21963238) or RAP (PMID: 23828888).

Response1: We completely agree with the suggestion of Reviewer 1 regarding the analysis of LINC00941 association with MTA2-occupied chromatin regions - this experiment would provide a valuable, additional verification of our hypothesis. Therefore, we performed a ChIRP experiment targeting endogenous LINC00941 with sets of eight specific DNA oligonucleotides tiling the lncRNA sequence, as well as a probe set targeting LacZ as a negative control. Upon analysis of the DNA sequencing results, we found that only one of the four replicate samples with LINC00941 pulldown yielded sufficient dynamic range in chromatin binding and we therefore had to discard the other three samples due to technical reasons. However, the remaining LINC00941 ChIRP sample (probe set "even 4") showed a clear enrichment of LINC00941 occupancy within chromatin regions bound by MTA2, when compared to our negative control (LacZ) as well as input samples. Importantly, LINC00941 was also enriched within the EGR3 locus. Of interest, this enrichment seems to specifically overlap with the MTA2 occupancy peaks detected by our MTA2 ChIP sequencing analysis.

[Figures removed by editorial staff per authors' request]

Clearly, these data cannot be included in the manuscript because of the lack of replicate LINC00941 pulldown samples. However, the ChIRP results show LINC00941 occupancy at MTA2-associated sites - including the EGR3 gene locus- and therefore are completely in line with our proposed hypothesis. Unfortunately, and honestly, we will not be able to repeat this experiment as part of the requested time for the revision process. Because of my recent transition from University of Regensburg to MSH Medical School Hamburg, we are currently in the process of moving our laboratory procedures and expertise to the new site. Therefore, we do currently not have the capacities to repeat this extremely time-consuming and financially straining analysis, which will require 480 to 720 million primary human keratinocytes to acquire sufficient amounts of material for detectable pulldown of endogenous LINC00941. We therefore hope, that the results we presented in the manuscript (including altered MTA2 chromatin occupancy in LINC00941-knockdown cells, phenocopy of increased

differentiation in MTA2- and LINC00941-deficient organotypic epidermis, LINC00941/MTA2-dependent repression of EGR3 expression and reduced differentiation in EGR3-deficient organotypic epidermis), together with the preliminary ChIRP result shown above will be sufficiently convincing to support our hypothesis.

Comment 2: In addition, the authors are also encouraged to address the following questions and comments to further improve the manuscript.

Fig 1C: Since multiple NuRD subunits were identified in the LINC00941 pull-down (Fig 1B), can the authors validate at least one other subunit? CHD4 is also a NuRD-specific subunit and appears to be a strong hit based on supplemental Fig 1.

Response 2: *We agree with Reviewer 1 that demonstrating the interaction between LINC00941 and CHD4 as another core component of the NuRD complex is desirable. Correspondingly, we performed RNA immunoprecipitation experiments targeting CHD4 and subsequent qRT-PCR analysis showed significant enrichment of LINC00941. These results can be found in Supplement Figure 1B of the revised manuscript.*

Comment 3: Fig 1D: Can the authors also try RNA-IP on MTA2 and endogenous LINC00941?

Response 3: *We thank Reviewer 1 for this comment. We have tried to perform the RNA-IPs with endogenous LINC00941, but due to the very moderate levels of this lncRNA in primary keratinocytes, it was technically not possible to achieve significant enrichment of lncRNA bound proteins above the general background level. Therefore, we had to do these experiments with slight overexpression of the lncRNA and performed the IPs with transduced LINC00941 in human primary keratinocytes. The amount of overexpression was carefully titrated so that it was moderately increased to about 10-fold compared with endogenous LINC00941 to avoid artificial effects.*

Comment 4: Fig 2B: It seems that MTA2 protein level still remains reasonably high at day-4.

Response 4: *We acknowledge the comment of Reviewer 1 as the data from the qRT-PCR analysis and Western Blot analysis might suggest a difference between MTA2 expression on RNA and protein level. Correspondingly, we repeated the Western Blot experiment with antibodies targeting MTA2, as well as CHD4 as an additional representative core subunit of the NuRD complex, and extended the timeline to six days of keratinocyte differentiation to allow for better comparability to our qPCR results (Figure 2A). Our results indicate a significant and continuous reduction of MTA2 as well as CHD4 protein abundance starting at day 1 of keratinocyte differentiation. The new data can be found in Figure 2b of the revised manuscript.*

Comment 5: Fig 3C: How many bivalent promoters are there in keratinocytes? How many of those are bound by MTA2?

Response 5: *We appreciate this question by Reviewer 1. In total, 19,994 protein-coding genes were tested. Out of these 7,653 genes are marked with bivalent promoter (chrom state Tss. Biv within +/- 1,000 bp distance to TSS) and include 1,085 genes bound by MTA2 as shown in the Venn diagram included in the revised manuscript (Supplement Figure 3D).*

Comment 6: Fig S3A: Can the authors examine MTA2 occupancy at TSS and bivalent TSS in control vs. siLINC00941 cells (by meta-gene analysis)? This will show whether LINC00941 KD affects MTA2 occupancy at bivalent TSS in general.

Response 6: *We appreciate the helpful suggestion of Reviewer 1 to further elucidate the intricacies of the LINC00941 and MTA2 relationship. Correspondingly, we conducted the meta-analysis of MTA2 occupied sites at bivalent promoter (overlap is presented in Venn diagram in Supplement Figure 3D).*

We found no significant differences between siControl and siLINC00941 treated cells in the meta-analysis (Supplement Figure 4C). This observation suggests that the knockdown of LINC00941 does not globally influence MTA2 binding at bivalent promoters and that the effects of LINC00941 are more selective.

Comment 7: Fig 4B: Does LINC00941 KD only affect 33 out of the 3613 MTA2 peaks? If yes, can the authors comment on why only such a small fraction of MTA2 occupied regions are affected?

Response 7: We thank Reviewer 1 for this question. The observation that only 33 out of 3613 MTA2 peaks passed our set criteria in Figure 4B can be explained through several factors. First of all, we used a False Discovery Rate of 5% and employed 3 biological replicates. As this setting is reliable for identifying clear changes in MTA2 occupancy, positions with more subtle changes might be missed. In addition, siRNA-mediated knockdown does not yield 100% depletion of the targeted RNA. Correspondingly, cells treated with siRNA pools targeting LINC00941 retained a certain amount of the lncRNA (Figure 2D, Supplement Figure 2B), which might maintain part of its functional effects on MTA2 occupancy. Moreover, DNA binding is a multifaceted process influenced by multiple factors. It's conceivable that while LINC00941 affects a subset of MTA2 regulated sites, others might be influenced by different factors (potentially even other lncRNAs), which were not under modulation in our experiment. Therefore, only a fraction might respond to a specific perturbation (Supplement Figure 4B; Supplement Figure 4C). Such specificity can be crucial for fine-tuned biological processes in epidermal differentiation where only certain pathways or processes need to be modulated. While at first glance it might seem like a small fraction, it indicates precision and specificity of the molecular processes under LINC00941 regulation. We included this important point in the discussion section of our revised manuscript.

Comment 8: Fig 4C: The authors only examined a small number of MTA2-associated genes. To provide a more complete view of the potential involvement of LINC00941-regulated genes in keratinocytes differentiation, can the authors provide the total number of differentially expressed genes (DEGs) in LINC00941 KD, the total number of DEGs during keratinocytes differentiation, and the overlap between the two (maybe using a venn diagram)? In addition, among all the overlapping DEGs from above, how many of them have MTA2 peaks nearby? Finally, in the overlapping DEGs occupied by MTA2, can the authors compare MTA2 occupancy at up- vs. down-regulated DEGs caused by LINC00941 KD, to see whether reduced MTA2 occupancy associates with increased expression after LINC00941 KD?

Response 8: We appreciate the suggestion of Reviewer 1 to further elucidate the intricacies of the LINC00941 and MTA2 relationship. Therefore, we conducted the meta-analysis of MTA2 occupied sites at siLINC00941 up- and down-regulated genes which are also differentially expressed in keratinocyte differentiation and have MTA2 bound at the promoter. However, we found no significant differences between siControl and siLINC00941 treated cells in the meta-analysis of MTA2 occupancy (Supplement Figure 4B). This observation suggests that the knockdown of LINC00941 does not globally influence MTA2 binding and the effects of LINC00941 are more selective. Here we would like to emphasize that we considered all MTA2 sites found in our analysis for the differential MTA2 binding analysis. Figure 4B shows those MTA2 sites, which were significantly altered as defined by an FDR of 5% cutoff.

Comment 9: Fig 4D: Can the authors add the H3K4me3 track to the figure? Can the authors provide ChIP-qPCR result to validate the changes in MTA2 occupancy near EGR3 after LINC00941 KD?

Response 9: We thank Reviewer 1 for this suggestion. Unfortunately, there are no corresponding H3K4me3 ChIP sequencing data available for the time course of epidermal differentiation provided by the Encode consortium. However, we performed ChIP-qPCR analyses as a validation of MTA2

occupancy changes at the EGR3 locus upon LINC00941 knockdown. These results can be found in Fig. 4E of the revised article.

Comment 10: Fig 5A: Some of the EGR3 target genes (eg., GJB4, SERTAD1) appear to be expressed before EGR3 up-regulation in siCtrl, and some of them (eg. HMOX1, ESYT3, SMPD3) appear to show stronger up-regulation than EGR3 in siLINC00941. This is not entirely consistent with the idea that they are regulated by LINC00941 via EGR3.

Response 10: We thank Reviewer 1 for this comment. We agree that the observation that some EGR3 target genes display stronger up-regulation than EGR3 itself in the siLINC00941 context. However, we believe that this does not negate the proposed regulatory model. The strength of regulator gene expression does not necessarily reflect the strength of target gene expression. Factors, such as amplification effects in downstream genes, or the presence of auxiliary regulatory mechanisms can lead to such observations.

As for the expression of some target genes prior to EGR3 up-regulation in siCtrl, it is essential to refer back to the original classification in the publication by Kim et al. Genes were classified as EGR3 targets based on specific criteria set in that study (EGR3 bound and upregulated genes, Co-expressed genes with EGR3 and mRNAs increased after 7d of Ca treatment), which may not fully overlap with the dynamics observed in our experimental setup. Our experimental procedures differ from the one in the Kim et al. study. We employed a organotypic tissue regeneration model, which can introduce a set of dynamic changes in gene expression that might not be present or as pronounced in the original study using 2D differentiation systems. This variation can account for some of the observed discrepancies. It's worth emphasizing that the role of EGR3 in the regulation of epidermal differentiation was further validated in an organotypic epidermal tissue context, as depicted in Figure 5C-D. Despite the differences in the experimental systems used between our study and Kim et al., it's noteworthy that there's a substantial overlap in the gene expression patterns and identified EGR3 targets. This overlap underscores the consistency and potential importance of the identified regulatory network.

In summary, while we acknowledge the discrepancies noted by the reviewer in Figure 5A, the entirety of the data—spanning different systems and validation experiments—supports the idea that EGR3 target genes are influenced by LINC00941 via EGR3. However, the nuances in gene expression dynamics emphasize the complexity of regulatory networks and the influence of experimental conditions.

Reviewer #1 (Significance (Required)):

The interaction between LINC00941 and NuRD is a novel finding and will likely provide new insights for the function of LINC00941, which has been implicated in keratinocytes, tissue homeostasis and cancer. It will also shed light on the role of lncRNAs in epigenetic gene regulation and cell fate transition in general. The conclusion of this study can be much strengthened if the authors can identify LINC00941-occupied genomic regions by ChIRP (PMID: 21963238) or RAP (PMID: 23828888). In addition, the authors are also encouraged to address the above-mentioned questions and comments to further improve the manuscript.

Reviewer #2 (Evidence, reproducibility and clarity (Required)):

In this manuscript, the authors investigate the role of LINC00941 epidermal differentiation. Specifically the authors show interaction with MTA2 and other NuRD subunits. Next, the authors show that LINC00941 and MTA2 restricts premature keratinocyte differentiation, where KD of either results in increased differentiation marker expression. To understand molecular impacts, the authors perform ChIPseq of MTA2 in control and LINC00941 depletion. Curiously, MTA2 binds in a trend differing from other cell types with predominant binding over active promoters. Upon LINC00941 KD,

MTA2 binding is changed at 33 locations, where the majority show reduced binding. Overlapping binding changes with gene expression changes, the authors identify EGR3 as the only direct candidate upregulated upon LINC00941 KD and upregulated during differentiation. KD of EGR3 results in opposite trends of LINC00941 KD, suggesting the proposed mechanism of LINC00941 repressing EGR3 until appropriate time in differentiation. I have the following suggestions for this work:

1) While data support MTA2 acting in NuRD, beyond Fig 1, the authors exclusively use MTA2 as a proxy for NuRD. Of course there are some subunits that are within other complexes and should not be used, others are options. While I do not expect the authors to perform all experiments on an additional subunit of NuRD, I do think there are a few things the authors should consider:

a. Be more precise with language to point out only MTA2 rather than say NuRD complex throughout many aspects of the paper, and only assume the complex in limited settings and when it is clear it is speculative

Response 1a: We thank Reviewer 2 for this suggestion and agree with the necessity of a careful usage of the term NuRD complex only when speculating or if direct proof for the involvement of the complex is present. For this reason, we have specifically mentioned MTA2 - and not NuRD- whenever involvement of the NuRD complex was likely, but not clearly proven. This includes the RNA-IP experiment to verify interaction between LINC00941 and MTA2 or CHD4, and also the MTA2 ChIP sequencing. In addition, we changed the wording throughout the manuscript to more precisely refer to MTA2 instead of NuRD wherever appropriate. Interestingly, RT-PCR analyses with LINC00941 knockdown keratinocytes showed significantly decreased abundance of MTA1, MTA3, CHD3 and CHD4 mRNAs upon LINC00941 and MTA2 knockdown, respectively This newly acquired data suggests impairment of NuRD complex formation upon knockdown of LINC00941 or associated MTA2, respectively and therefore indicates that the lncRNA might indeed be associated with the NuRD complex. In addition, we performed RNA immunoprecipitation experiments with an antibody targeting CHD4 and found significant enrichment of LINC00941. These results further strengthen our hypothesis of LINC00941 association with the NuRD complex and can be found in the new version of Supplement Figure 1B (RNA-IP) and Supplement Figure 2B-C (RT-PCR analyses) of the revised manuscript.

b. Perform a subset of experiments on another subunit. For example, the Mass Spec in Fig 1A/B shows an interaction with other subunits, but the verification was only done for MTA2 (Fig 1C/D). This could easily be blotted (or another Western performed) and/or primers for other subunits for the qPCR for a couple additional subunits. Similarly straightforward, looking at MTA2 RNA expression changes during differentiation (Fig 2A): if additional primers were used to other subunits, these additional subunits could be used to verify.

Response 1b: We agree with Reviewer 2 that demonstrating the interaction between LINC00941 and CHD4 as another core component of the NuRD complex is desirable. Correspondingly, we performed RNA immunoprecipitation experiments targeting CHD4 and subsequent qRT-PCR analysis showed significant enrichment of LINC00941. These results can be found in Supplement Figure 1B of the revised manuscript. In addition, we performed qRT-PCR analyses to measure CHD4 mRNA abundance during keratinocyte differentiation and could show that CHD4 mRNA levels decreased during the course of keratinocyte differentiation. This new data is included Fig. 2A of the revised version of our article. Furthermore, we carried out Western Blot analyses to analyze dynamic regulation of CHD4 protein levels during keratinocyte differentiation. In accordance with our hypothesis, CHD4 protein abundance is highest in undifferentiated keratinocytes and drastically decreases upon onset of calcium-induced differentiation (Figure 2b of our revised manuscript).

2) Related to the above comment, does MTA2 KD (or LINC00941 KD for that matter) result in loss of NuRD complex formation? If so, this would be sufficient to address point 1.

Response 2: We thank Reviewer 2 for this interesting suggestion. To test this hypothesis, we performed qRT-PCR analysis of other NuRD components (MTA1, MTA4, CHD3, CHD4) to detect their abundances in MTA2 knockdown and LINC00941 knockdown keratinocytes, thus gaining an indication of whether NuRD even forms at all in these cells. Interestingly, MTA1, MTA3, CHD3 and CHD4 show significantly decreased mRNA abundance upon MTA2, as well as LINC0941 knockdown, respectively. This data strongly indicates impairment of NuRD complex formation upon knockdown of LINC00941 or associated MTA2 and can be found in Supplement Figure 2B-C of the revised manuscript.

3) Finally, in relation to NuRD complex here, it is important to note that mutually exclusive NuRD complexes (MBD2/NuRD and MBD3/NuRD) have been documented. Because the Mass Spec did not show interaction to MBD2 or MBD3, it is not clear if this is limited to one of these complexes. Related to this, the authors show by Mass spec that LINC00941 interacts with CHD4, but not CHD3. Is this because Chd3 is not expressed in these cells, or because there is some mutual exclusivity to CHD4 and LINC00941 is acting through this subcomplex?

Response 3: We thank Reviewer 2 for this question. We performed the Mass Spec experiments several times with slightly varying washing steps. The data presented in this manuscript are from the experimental approach with the most stringent washing steps. However, other Mass Spec experiments with less stringent washing procedures (10 mM NH₄HCO₃/acetonitrile (1/1) instead of 50 mM NH₄HCO₃/acetonitrile (1/1) also detected the MBD2 and MBD3 subunits of the NuRD complex. Therefore, we assume that these molecules also interact at least indirectly with LINC00941, but with lower efficiency, which is why they are lost during stringent washing steps. In case of CHD3 and CHD4, respectively, we suggest that only CHD4 is present in the NuRD complexes that interact with lncRNAs, since so far only this isoform has been shown to interact with ncRNAs. A simultaneous occurrence of CHD3 can therefore be excluded because stoichiometric analysis has shown that only one of these two isoforms is present simultaneously in the NuRD complex (Low et al., 2020).

4) Immunofluorescence images showing increased Keratin 10 and Filaggrin in LINC00941 or MTA2 KD (Fig 2E) and decreased Keratin 10 and Filaggrin in EGR3 KD (Fig 5C) are curious as the control look very different. In 2E, the control shows barely detectable levels, whereas in 5C the levels look similar to what is seen in Fig 2E KDs. Is this variability? If so, more representative images as well as quantification to the changes are necessary to make these two points.

Response4: We thank Reviewer 2 for this comment with respect to different signal intensities between immunofluorescence images. Variances between Figure 2E and Figure 5D (Figure 5C in the original version) originate from deviating exposure times. While all exposure times within each experiment were kept the same between all tissues analyzed, the immunofluorescence images compared by the Reviewers come from different experiments that were not performed at the same time, and where knockdown yielded drastically different changes in abundance of Keratin10 and Filaggrin proteins. Correspondingly, due to the strong increase of K10 and FLG protein abundances at MTA2 and LINC00941 knockdown tissues, respectively, exposure times were kept short for this experiment, to avoid overexposure - which resulted in weak visibility of the respective signals in control tissues. On the other hand, exposure times were increased in the EGR3 experiment because of a strong reduction of K10 and FLG protein levels in EGR3 knockdown tissues - thus resulting in higher visibility of the signals in the respective control tissues of that experiment. Importantly, all control-treated tissues ("siControl") of a given experiment were always recorded with identical exposure times to the corresponding siPool-treated knockdown tissues. In addition, all organotypic epidermis cultures of all treatment groups within experiments were harvested at the same developmental stage and time point.

5) In Figure 4, the authors present ChIPseq data for MTA2 in LINC00941 KD. One interesting trend is that the KD alters binding of MTA2 at mostly bivalent/repressed locations, rather than at active

locations which is the majority of MTA2 binding (from Fig 3). It would be nice to show then these data rather than only stating it. The authors include a browser track for 2 genes (Fig 4D and S4C), but for the other 31 locations, a heatmap or something to show the level of K27me3 vs K27ac/K4me3 would be helpful to support this claim. Notably saying "Most of the differential MTA2/NuRD occupied sites were marked by repressive histone modification H3K27me3..." is the point that doesn't seem to be shown, and also a precise number should be included.

Response 5a: We thank Reviewer 2 for this suggestion. Here, we used the pre-defined 15-state chromatin model of male foreskin keratinocytes (roadmap accession: E057), which is based on the integration of multiple ChIP-seq data of various histone modifications through a multivariate Hidden Markov Model, chromHMM (Ernst & Manolis, 2017). This is to our knowledge widely used and the most accurate prediction of chromatin states. However, in order to provide more clarity regarding the presentation of data, we have made the following changes. In Figure 4B, we have replaced the term "chromHMM" with "chromatin state". Furthermore, we added Supplemental Material 3, which includes differential MTA2 peaks upon LINC00941 knockdown. The column "main.chromState" reports the main chromatin state of each MTA2 occupied region that significantly changed.

a. Related to this, I believe the authors performed K27me3 ChIPseq in the KD, and if so, it would be nice to see more genome wide effects here.

Response 5b: We thank Reviewer 2 for this comment. However, we did not perform K27me3 ChIP sequencing, but only MTA2 ChIP sequencing to show the dependency of MTA2 chromatin binding behavior on the presence or absence of LINC00941. The ChIP sequencing tracks of histone modification shown in the manuscript were derived from publicly available data from the Encode consortium, as stated in the corresponding section of "Materials and Methods".

6) This is perhaps beyond the scope of the paper, but the obvious question to me is if EGR3 is relocalized in LINC00941 KD. Specifically, we would anticipate that EGR3 localization in the KD would mimic that of a more differentiated cell (binding to differentiated genes). A quick ChIPqPCR experiment for a few locations would be sufficient to support this model.

Response 6: We thank Reviewer 2 for this suggestion and agree that an in-depth analysis of dynamic EGR3 occupancy in the course of keratinocyte differentiation of control and LINC00941 knockdown cells would be of interest. However, since we work with primary human keratinocytes which are highly transfection-resistant and only usable for a limited number of passages, our only way of efficiently introducing siRNA pools is via electroporation - a procedure that results in a significant number of cells that don't survive the procedure. Therefore, an adequately controlled experiment to test knockdown and control cells in non-differentiated and several differentiated conditions (triplicate samples) would require approximately 1.3 billion primary cells - no matter if we would perform DNA sequencing or qPCR analyses with the resulting ChIP samples. This assumption is based on our MTA2-ChIP sequencing experiment included in this manuscript, which was performed with siControl and siLINC00941 primary keratinocytes and only involved non-differentiated cells. Due to the very large scope of this experiment, we will not have the capacities to provide these results as part of the revision process.

Minor points:

1) Importantly, CHD5 can also be incorporated in NuRD, in place of CHD3 or CHD4.

Response Minor 1: We thank Reviewer 2 for this suggestion. Our previous sequencing analysis of organotypic epidermis (Graf et al., 2019) has shown that CHD5 is not significantly expressed in human keratinocytes, thus, we did not mention this NuRD subunit in our manuscript.

2) The authors use heatmaps and metaplots in Fig S3 to show reproducibility of the ChIPseq datasets. Importantly, the PCA does show some variation. XY scatterplots for replicates vs one another would be a more robust QC.

Response Minor 2: We thank Reviewer 2 for this comment. However, as PCA is a technique that transforms high-dimensions data into lower-dimension while retaining as much information as possible we believe that PCA is not suited to make any statement about the absolute variation of the data, because it represents the relative variation within the data and tries to maximize it into a few dimensions. As the PC1 and PC2 shown are scaled to the variance they represent, we cannot directly compare the values shown on x- and y-axis. We would expect that biological replicates are not identical but show some variation due to experimental limitations. Therefore, it is not surprising that we see variation between replicates on PC2, as stated by the reviewer. However, PC1 clearly shows the separation between siRNA conditions and accounts for way higher variance in the data (PC1 50% and PC2 15%), confirming that the effect of the knockdown is the main cause of variation in the data. However, we added the requested scatterplots to the supplement figures to underscore the high reproducibility of our replicates.

3) In figure 4D, the authors present nice data showing changes in histone mods during differentiation, but it is very hard to see the color changes and the tracks as presented. (same point for Fig S4C)

Response Minor 3: We thank Reviewer 2 for highlighting this point. For better clarity, we have revised the color scheme of these Figures.

4) it is unclear from the methods or the figure legend if RTqPCR data are biological or technical replicates.

Response Minor 4: The authors thank Reviewer 2 for this comment. We updated the section "qRT-PCR analysis" (Materials and Methods) to clarify that we used biological replicates.

Reviewer #2 (Significance (Required)):

In this manuscript, the authors present a molecular function for LINC00941 in epidermal differentiation, where it interacts directly with NuRD subunit MTA2. LINC00941 has been previously described but this activity was not described. LINC00941 seems to specifically help target or maintain MTA2 localization to EGR3 to promote repression of this gene. Then, the model suggests that during differentiation, LINC00941 and MTA2 levels decrease, permitting activation of EGR3 during epidermal differentiation and subsequent activation of appropriate genes. These findings will be of interest to individuals interested in NuRD function, lncRNA activity and/or epidermal cell fate. My expertise is in chromatin biology, chromatin remodelers, epigenomics, and cell identity.

Reviewer #3 (Evidence, reproducibility and clarity (Required)):

In the manuscript entitled "The long non-coding RNA LINC00941 modulates MTA2/NuRD occupancy to suppress premature human epidermal differentiation", Morgenstern and colleagues describe a novel mechanism by which Nur complex interacting with a ncRNA LINC00947 represses expression of a late differentiation transcription factor EGR3 in the skin. These findings are novel and will be of interest in the field. Of note, the authors use a plethora of biochemical and cell biology techniques

such as chromatin occupancy, transcriptomics and organotypic skin culture. Both the Introduction and Discussion outline a necessary background in the field of ncRNAs and skin biology as well as clearly state and discuss the scientific problem. The Methods are comprehensive and accurate. Importantly, any possible ethical issues are discussed. The Results are clear and support the data shown. Also the Figures are well organised and easy to follow. Overall, the manuscript is well prepared and will be of interest for broad readership. Before publication, however, the authors should address some points to further enhance their work:

Major points:

1. It is peculiar that there is no evident K10 and FLG staining in the organotypic skin from siControl cells in Fig 2E but it is present on the Fig 5C. The authors should explain these inconsistencies. Ideally the authors should provide a western blot analysis of these proteins which would help to give a more quantitative picture of the phenomenon.

Response 1: We thank Reviewer 3 for this comment with respect to different signal intensities between immunofluorescence images. Variances between Figure 2E and Figure 5D (Figure 5C in the original version) originate from deviating exposure times. While all exposure times within each experiment were kept the same between all tissues analyzed, the immunofluorescence images compared by the Reviewers come from different experiments that were not performed at the same time, and where knockdown yielded drastically different changes in abundance of Keratin10 and Filaggrin proteins. Correspondingly, due to the strong increase of K10 and FLG protein abundances at MTA2 and LINC00941 knockdown tissues, respectively, exposure times were kept short for this experiment, to avoid overexposure - which resulted in weak visibility of the respective signals in control tissues. On the other hand, exposure times were increased in the EGR3 experiment because of a strong reduction of K10 and FLG protein levels in EGR3 knockdown tissues - thus resulting in higher visibility of the signals in the respective control tissues of that experiment. Importantly, all control-treated tissues ("siControl") of a given experiment were always recorded with identical exposure times to the corresponding siPool-treated knockdown tissues. In addition, all organotypic epidermis cultures of all treatment groups within experiments were harvested at the same developmental stage and time point.

2. There are no evidences at protein level of an efficiency of MTA2 and EGR3 knock-down. The authors use an MTA2 for western blot, so it should not be difficult to confirm.

Response 2a: We thank Reviewer 3 for this comment. To clarify this issue, we carried out Western Blot analyses with protein lysates of organotypic epidermis cultures generated with siControl-, siEGR3-, and siLINC00941- treated keratinocytes, respectively. The results clearly show that siPool-mediated knockdown of EGR3 led to a decrease of EGR3 protein abundance, while LINC00941-deficiency resulted in the expected increase of EGR3 protein levels - in a similar fashion as previously seen on RNA level. These new results can be found in the revised Supplemental Figure 5B).

Analogously, we verified efficient knockdown of MTA2 protein in primary keratinocytes treated with siRNA-pools targeting MTA2 via Western blot, as well as in MTA2 knockdown organotypic epidermis with the help of immunofluorescence analyses (revised Supplemental Figure 2A).

For EGR3, it is essential, as EFR3 is expected to increase only during late differentiation, while experiments from Fig 5B are performed only at 3 days of differentiation. Is it enough to induce EGR3 expression? Is the knock-down efficient at protein level at that time point? This is important as the work by Kim et al 2019 shows a detectable expression of EGR3 only after 7 days of differentiation.

Response 2b: We thank Reviewer 3 for raising this important point. It is important to note, that the model systems used by Kim et al. (2019) are substantially different from the ones used in this study: To our knowledge, the experiment Kim et al. used to detect onset of EGR3 expression after 7 days of differentiation was performed in cultured keratinocytes. We analyzed the functional effects of EGR3-mediated knockdown (shown in Figure 5C-D) in organotypic epidermis tissue. Since the dynamics of epidermal regeneration -including establishment of a non-differentiated, progenitor cell-containing basal layer and epidermal stratification + terminal differentiation are very different from calcium-induced differentiation of keratinocytes in a 2D model, we do not expect similar dynamics in EGR3 expression between both studies. In addition, our organotypic tissue model does not require initial culturing of keratinocytes in submerged conditions and therefore shows protein expression of late differentiation genes, such as Filaggrin (Figure 5D), as well as development of late differentiation ultrastructure already at day 3. Also, we observed robust EGR3 mRNA abundance in our organotypic epidermis at day 3. However, we completely agree with the Reviewer, that analysis of EGR3 protein abundance in our model systems will clearly help to verify efficient and functionally relevant reduction of EGR3 protein abundance upon knockdown. Correspondingly, we performed Western blot in undifferentiated primary keratinocytes compared to keratinocytes throughout six days of differentiation. As a result, we observed the expected very low levels of EGR3 protein in undifferentiated keratinocytes and continuous increase in abundance throughout differentiation (revised Figure 5B). Importantly, we could not only detect robust levels of EGR3 protein on day three of keratinocyte differentiation, but also in our three days old organotypic epidermis model, which also shows significant EGR3 protein knockdown upon siEGR3-treatment (revised Supplemental Figure 5B).

3. The authors should demonstrate an increase of EGR3 at protein level after LINC00947 knock-down (by western blot in vitro and/or in epidermal organotypic tissue).

Response: We thank Reviewer 3 for this helpful suggestion. As mentioned above, we carried out Western Blot analyses with protein lysates of organotypic epidermis cultures generated with siControl-, siEGR3-, and siLINC00941- treated keratinocytes, respectively. The results clearly show that siPool-mediated knockdown EGR3 led to a decrease of EGR3 protein abundance, while LINC00941-deficiency resulted in the expected increase of EGR3 protein (revised Supplemental Figure 5B).

4. A key experiment to confirm the proposed model should be a double knock-down of both LINC00941 and EGF3. Will it rescue the observed increase of pro-differentiation genes?

Response: We thank Reviewer 3 for this suggestion. As requested, we generated organotypic epidermal tissue with keratinocytes simultaneously treated with EGR3- as well as LINC00941 siRNA pools. Interestingly, we observed no reduction, but a moderate increase of EGR3 mRNA levels in siEGR3 + siLINC00941 double-treated tissue. These results suggest, that the siRNA-mediated posttranscriptional knockdown was not efficient enough to completely abrogate the increase in EGR3 transcription caused by knockdown of LINC00941. Correspondingly, differentiation mRNA abundance (FLG, LCE-1E, KRT1) was increased, albeit not as highly induced as seen in LINC00941 single knockdown tissue (Figure 2D). While full knockout of EGR3 and LINC00941 would be required to observe stronger effects, this experiment verifies the functional interdependency of LINC00941 and EGR3 and thus supports our hypothesis of LINC00941/NuRD-mediated regulation of EGR3 transcription.

Minor points:

1. 4E - EGR3 or LINC00941 knock-down?

Response Minor 1: We acknowledge the comment of Reviewer 3 and corrected Figure 4F (Figure 4E in the original version of the manuscript) (LINC00941 knockdown instead of EGR3 knockdown).

2. 4E - The RNA seq label states "RNA seq (siLINC00941 d3)" but apparently shows both scr and siLINC00941

Response Minor 2: We thank Reviewer 3 for this comment and have revised Figure 4D accordingly.

3. Please add the recipe of the lysis buffer used for western blot analysis of keratinocytes lysates.

Response Minor 3: The authors thank Reviewer 3 for this comment. We have included the lysis buffer recipe in the "Western Blot analysis" section (Materials and Methods).

4. Page 10 - Fig 4D description - is it "chromatin conformation state" or just "chromatin state"?

Response Minor 4: We revised the corresponding passages from "chromatin conformation state" to "chromatin state".

Reviewer #3 (Significance (Required)):

These findings are novel and will be of interest in the field.

April 3, 2024

RE: Life Science Alliance Manuscript #LSA-2023-02475R

Dr. Markus Kretz
Medical School Hamburg
Am Kaiserkai 1
Hamburg 20457
Germany

Dear Dr. Kretz,

Thank you for submitting your revised manuscript entitled "LncRNA LINC00941 modulates MTA2/NuRD occupancy to suppress premature human epidermal differentiation". We would be happy to publish your paper in Life Science Alliance pending final revisions necessary to meet our formatting guidelines.

- please be sure that the authorship listing and order is correct
- please add the Twitter handle of your host institute/organization as well as your own or/and one of the authors in our system
- please note that the titles in the system and on the manuscript file must match
- please separate the Figure legends and Supplemental Figure legends into separate sections
- please add callouts for Figures 3D; 4E and S5C to your main manuscript text
- please make datasets publicly accessible at this point, and update accession information in the Data Availability statement

FIGURE CHECKS:

- please add sizes next to all blots

A. FINAL FILES:

B. MANUSCRIPT ORGANIZATION AND FORMATTING:

Sincerely,

Reviewer #1 (Comments to the Authors (Required)):

The authors have addressed all my concerns.

Reviewer #2 (Comments to the Authors (Required)):

The authors have addressed questions and comments raised in the previous round of review and the manuscript has been significantly strengthened.

Reviewer #3 (Comments to the Authors (Required)):

The authors addressed my concerns. The manuscript is now more solid and clear.

April 10, 2024

RE: Life Science Alliance Manuscript #LSA-2023-02475RR

Dr. Markus Kretz
Medical School Hamburg
Am Kaiserkai 1
Hamburg 20457
Germany

Dear Dr. Kretz,

Thank you for submitting your Research Article entitled "LncRNA LINC00941 modulates MTA2/NuRD occupancy to suppress premature human epidermal differentiation". It is a pleasure to let you know that your manuscript is now accepted for publication in Life Science Alliance. Congratulations on this interesting work.

DISTRIBUTION OF MATERIALS:

Again, congratulations on a very nice paper. I hope you found the review process to be constructive and are pleased with how the manuscript was handled editorially. We look forward to future exciting submissions from your lab.

Sincerely,
